# Let's disagree to agree: Evaluating collective disagreement among AI vision systems

## Abstract

Recent advancements in artificial intelligence (AI) have led to the development of AI vision systems that closely resemble biological vision in terms of both behavior and neural recordings. While prior research in modeling biological vision has largely concentrated on comparing *individual* AI systems to a biological counterpart, our study instead investigates the collective behavior of model populations. We focus on inputs that generate the most divergent responses among a diverse population of AI vision systems, as measured by their aggregate disagreement. We would expect that the factors driving disagreement among AI systems are also causes of misalignment between AI systems and human perception. We challenge this expectation by demonstrating alignment between AI systems and humans at the *population* level, even for images that generate divergent responses among AI systems. This unexpected finding challenges our understanding of the relationship between the limitations of AI systems and human perception, suggesting that even the most challenging stimuli for AI systems are reflective of human perceptual difficulties.

## 1 Introduction

As artificial intelligence (AI) systems scale in complexity, they tend to exhibit increasingly similar behavior and representations (Li et al., 2015; Geirhos et al., 2021; Huang et al., 2021; Sorscher et al., 2022), making it challenging to differentiate the unique computational properties of individual systems (Geirhos et al., 2018; Maheswaranathan et al., 2019; Han et al., 2023). Representational *convergence* or *universality* is a natural consequence of standard machine learning training because optimization of overparameterized systems on vast datasets leads to similar solutions despite implementational differences (Cao & Yamins, 2021; Huh et al., 2024; van Rossem & Saxe, 2024), with behavioral convergence as a downstream consequence. However, despite convergence in the aggregate, AI systems can disagree in their predictions for specific visual inputs, especially for artificial stimuli produced from corruptions poorly represented in their training diet (Geirhos et al., 2020).

One might expect that the stimuli driving the most disagreement among AI systems are those that cause these systems to deviate the most from human perception, with the intuition that these "disagreeable" images are challenging cases in which AI systems struggle to reach a consensus due to limitations in their training, architecture, or underlying assumptions about visual processing. This expectation aligns with standard approaches for comparing internal representations of AI and biological vision, such as representational similarity analysis (Kriegeskorte et al., 2008), which prescribe computing similarities between model and biological activations aggregated over a large-scale and often undifferentiated stimulus set. Moreover, this expectation is explicit in behavioral extrapolation tests, which exploit intrinsic properties of AI systems to degrade their alignment to human behavior, for example, translation invariance or sensitivity to adversarial perturbations (Geirhos et al., 2019; Madan et al., 2020; Hendrycks et al., 2021; Ollikka et al., 2024).

In this work, we challenge the assumption that disagreement among AI systems is intrinsic to these systems and unrelated to aspects of human visual processing. Instead, we aim to demonstrate that disagreement among populations offers a valuable opportunity for comparative analysis between

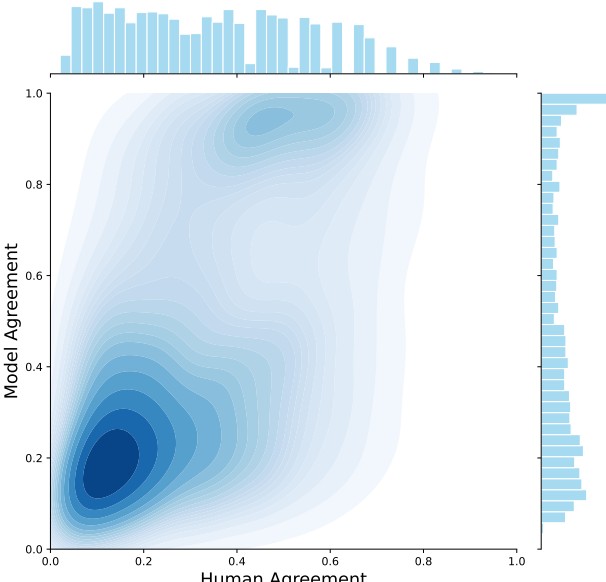

**Figure 1: (Dis)agreement among AI vision systems is correlated to (dis)agreement among humans.** A kernel density estimate plot of ObjectNet (Barbu et al., 2019) images corresponding to human and AI agreement levels. The $y$-axis indicates the label agreement level per Eq. (2) among 1032 AI systems (see Section (2.2)); the $x$-axis indicates the label agreement level per Eq. (2) among 42 human participants (see Section (2.4)); histograms along each axis reflect the proportion of images at each marginal agreement level. We observe a positive correlation between model agreement and human agreement, which demonstrates that the stimuli that cause the most (dis)agreement among AI systems also cause the most (dis)agreement among humans.

artificial and human vision systems, as it accounts for the aggregation of individual differences in visual processing that may be elicited by properties of individual stimuli.

Our contributions are as follows:

— We quantify disagreement among a large and diverse population of AI vision systems, comprising an order of magnitude more systems than comparable studies comparing artificial and human vision (*cf.* Geirhos et al., 2021), and demonstrate a correspondence between model disagreement and human disagreement; see Fig. (1) alongside later sections.

— We investigate the properties of *naturalistic* stimuli that elicit the most disagreement among this model population, in contrast to prior studies that construct artificial stimuli with properties tuned for particular aspects of artificial visual processing (*cf.* Geirhos et al., 2019; Hendrycks et al., 2021).

— We provide evidence that disagreement among AI vision systems is driven by aspects of human visual perception, particularly image difficulty as quantified by human behavioral data, suggesting that individual differences in AI vision systems may reflect individual differences in human visual processing rather than suboptimalities in artificial vision.

In conducting our study, we aim to uncover the factors that make certain images "disagreeable" for *both* AI vision systems and humans, and ultimately, how this understanding can guide the development of more robust and human-aligned vision systems, a problem of significant current interest (Wichmann & Geirhos, 2023; Sucholutsky et al., 2023). Moreover, our study, and the converging evidence for behavioral and representational convergence in the field at large, call for a rethinking of model comparison away from the paradigm of one-to-many towards many-to-many comparisons between artificial and biological vision.

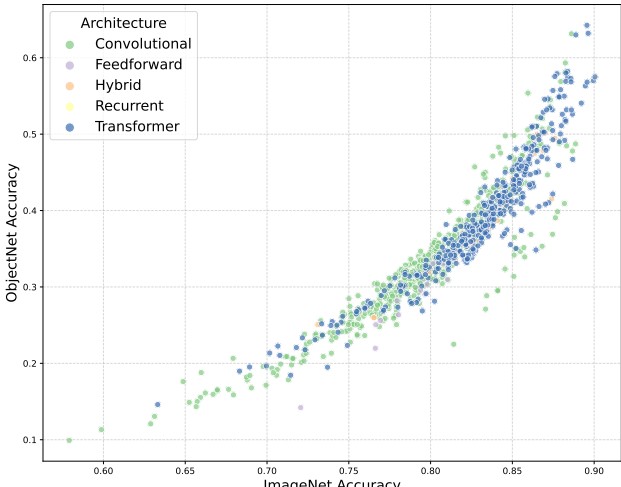

**Figure 2: Survey of the AI vision system population by architecture family.** Scatter plot comparing model performance (top-1 accuracy) on ObjectNet ($y$-axis) versus ImageNet ($x$-axis). Each point indicates a single model, with colors indicating its architecture family (convolutional, feedforward, hybrid, recurrent, or transformer). Higher ImageNet accuracy corresponds to higher ObjectNet accuracy, irrespective of architecture family, consistent with representational convergence.

## 2 METHODS

### 2.1 VISUAL STIMULUS SETS

We conduct our analyses on a diverse set of images drawn from ImageNet (Deng et al., 2009) and ObjectNet (Barbu et al., 2019), widely used benchmarks in computer vision research. ImageNet provides a dataset of labeled natural images across thousands of object categories, while ObjectNet introduces additional challenges by presenting objects in real-world scenarios with varied viewpoints and occlusions and has a "no training" policy to preserve its status as a test set. These datasets were chosen for their ability to test model performance across a wide range of object categories and difficulty levels, allowing us to capture a spectrum of disagreement. We use the 50 000 images among 1000 classes from the ILSVRC 2012–2017 validation set (Russakovsky et al., 2015) and the 50 000 images among 313 classes from the ObjectNet test set, for a total of 100 000 visual stimuli among 1200 classes due to the partial class overlap of ImageNet and ObjectNet. We will henceforth refer to these sets simply as "ImageNet" and "ObjectNet."

### 2.2 A POPULATION OF AI SYSTEMS

We examined the behavior of 1032 AI vision systems, spanning different architectures, including convolutional neural networks (CNNs), vision transformers (ViTs), and hybrid architectures; see Fig. (2) and Fig. (10). These systems varied in complexity, dataset size, and pre-training and fine-tuning protocols, representing a comprehensive cross-section of early and state-of-the-art visual recognition systems; see Appendix (A.1). We obtained the class label predictions of each system on ImageNet and ObjectNet, enabling us to analyze patterns of agreement and disagreement at scale.

### 2.3 MEASURING DISAGREEEMENT WITH FLEISS' $\kappa$

To quantify the level of agreement among the population of AI systems, we employ *Fleiss' $\kappa$* ("kappa"), a statistical measure used to assess the reliability of agreement between multiple raters assigning categorical ratings to a set of items (Fleiss, 1971). In our context, each AI system or human participant acts as a *rater* assigning a categorical *label* to each *stimulus* (image).

Formally, let $M$ represent the number of raters (AI systems or humans), $N$ the number of stimuli (images), $k$ the number of categories (class labels), and $T$ the prediction matrix of size $N \times k$, where

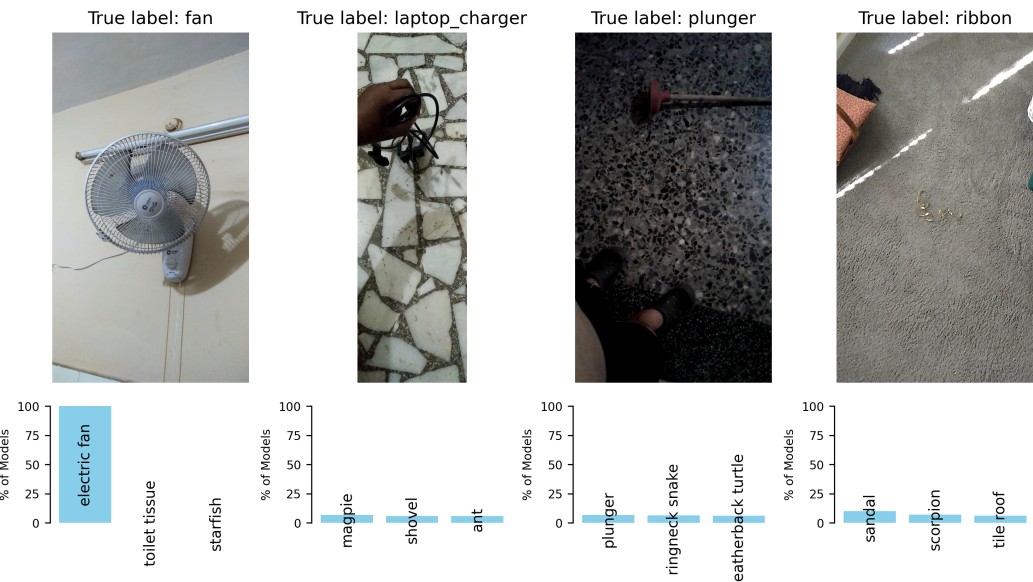

**Figure 3: One agreeable and three disagreeable ObjectNet images.** One image from ObjectNet that elicits the highest agreeement and three that elicit the highest disagreement (lowest per-stimulus agreement per Eq. (2)) amongst the population of vision models described in Section (2.2). Analogous images for ImageNet are provided in Appendix (A.5).

$T_{ij}$ represents the number of models that assign category $j$ to stimulus $i$. For each stimulus $i$, the total number of ratings is

$$n_i = \sum_{j=1}^{k} T_{ij} = M \ . \tag{1}$$

We compute the *per-stimulus agreement* as

$$p_{\text{agree},i} = \frac{\sum_{j=1}^{k} T_{ij}(T_{ij}-1)}{n_i(n_i-1)} = \frac{\sum_{j=1}^{k} T_{ij}(T_{ij}-1)}{M(M-1)} \ . \tag{2}$$

This value $p_{\text{agree},i}$ represents the extent to which the models agree on stimulus $i$, ranging from 0 (no agreement) to 1 (complete agreement). The *observed agreement* $P$ is the average of all per-stimulus agreements,

$$P = \frac{1}{N} \sum_{i=1}^{N} p_{\text{agree},i} \ . \tag{3}$$

We assume a uniform distribution over categories, which means the expected agreement by chance is $P_e = \frac{1}{k}$. Finally, Fleiss' $\kappa$ is computed as

$$\kappa = \frac{P - P_e}{1 - P_e} \ . \tag{4}$$

Eq. (4) quantifies the aggregate agreement among raters while adjusting for the agreement expected by chance, and ranges from $-1$ (complete disagreement) to 1 (perfect agreement), with 0 indicating no agreement beyond chance. Fleiss' $\kappa$ thus provides a measure of *inter-rater reliability* (agreement) among a *population* of raters (AI systems or humans) across a set of stimuli (images), and acts as a generalization of Cohen's $\kappa$, the chance-corrected measure of *pairwise* agreement between two systems most recently used by Geirhos et al. (2020) to compare individual AI systems to humans.

### 2.4 GROUNDING IN HUMAN BEHAVIORAL MEASURES

Mayo et al. (2023) collected a large dataset of human object recognition judgments consisting of 200 382 human responses from 2647 human participants for 4771 images, of which 2415 are from

ObjectNet and 2356 from ImageNet. Images were presented at 6 different durations between 17ms and 10s, with 42 responses collected per image (7 different subjects seeing each image at one of six timings). In the next section, we compare population-level agreement with two human perceptual measures derived from this dataset. The **minimum viewing time** is defined as the shortest duration at which the majority of human participants (more than half) can correctly recognize an object in an image. This metric serves as a proxy for image difficulty as longer times suggest that an image is more challenging for humans to interpret quickly. The **difficulty score** is the total number of incorrect responses out of the 42 presentations of each image. A higher difficulty score indicates that more humans struggled with the image.

ImageNet-X (Idrissi et al., 2022) provides annotations for almost all images in the ImageNet dataset (46 110 out of 50 000 images) across 16 factors of variation that capture how each image differs from prototypical examples of its class. Annotations were made by comparing each image to three class-prototypical images, with human inspection to categorize the differences into the 16 factors. The annotations cover factors of variation like pose, background, color, texture, size, lighting, and occlusion. While multiple factors could be selected, a *top factor* was also determined for each image to identify the main factor of deviation from the class prototypes. We take the intersection of images evaluated over models, humans and ImageNet-X annotated images which results in 2194 with agreement scores and annotations for both models and humans.

## 3 RESULTS

| Dataset | Minimum viewing time | Difficulty score | Human agreement |
|---------|:--------------------:|:----------------:|:---------------:|
| ImageNet | −0.29 | −0.33 | 0.30 |
| ObjectNet | −0.41 | −0.45 | 0.44 |

Table 1: **Correlation between model agreement and the human behavioral measures.** Per-stimulus model agreement (the $y$-axis of Fig. (1)) is negatively correlated with the human behavioral measures of per-image difficulty described in Section (2.4), and positively correlated with per-stimulus human agreement (the $x$-axis of Fig. (1)).

In this section, we investigate correspondences between model and human disagreement at the population level established in Fig. (1), as well as the human behavioral measures described in Section (2.4). As a summary, Table (1) displays the correlation coefficients between model agreement among AI vision models and the human behavioral measures across the two datasets, ImageNet and ObjectNet. Overall, the correlations are stronger (more negative or positive) for the ObjectNet dataset compared to ImageNet. ObjectNet contains images with more varied viewpoints, occlusions, and real-world complexities, making them more challenging for both humans and AI models. Examples of the lowest agreement images for ObjectNet are shown in Fig. (3).[1]

### 3.1 MINIMUM VIEWING TIME

The negative correlation in Table (1) means that as the minimum viewing time increases (*i.e.*, the image is harder for humans), the model agreement among AI systems decreases. In other words, images that require longer viewing times for humans tend to be the ones where AI vision systems disagree more. Fig. (4) demonstrates that many low-agreement images are associated with the longest viewing time (10 seconds). This suggests that both humans and AI vision systems find these images challenging, leading to lower agreement within both groups.

### 3.2 IMAGE DIFFICULTY

The negative correlation in Table (1) of agreement with difficulty score indicates that images with higher difficulty scores (more challenging for humans) correspond to lower model agreement among AI vision systems. This alignment implies that images difficult for humans are also difficult for AI vision systems, leading to greater disagreement among the models. Fig. (6) visually reinforces this

---

[1]Additional images from both low-agreement and high-agreement sets for both ImageNet and ObjectNet are available in Appendix (A.5).

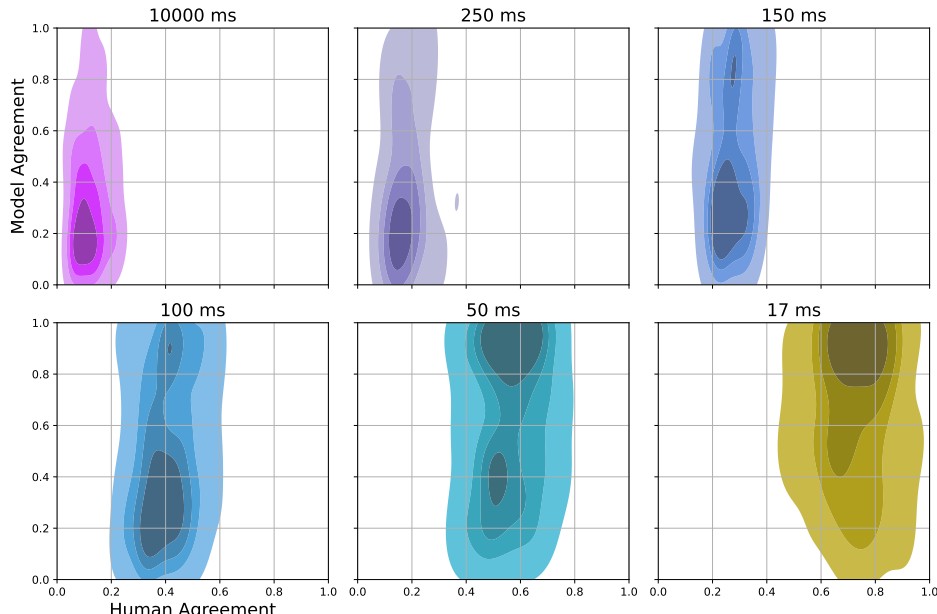

**Figure 4: Disagreeable images take more time to recognize.** Density estimate plots where images are binned into different minimum viewing times (minimum duration at which an image is recognized by a majority of humans (Mayo et al., 2023)). Human subjects tend to spend more time on images with low agreement among both human and AI systems.

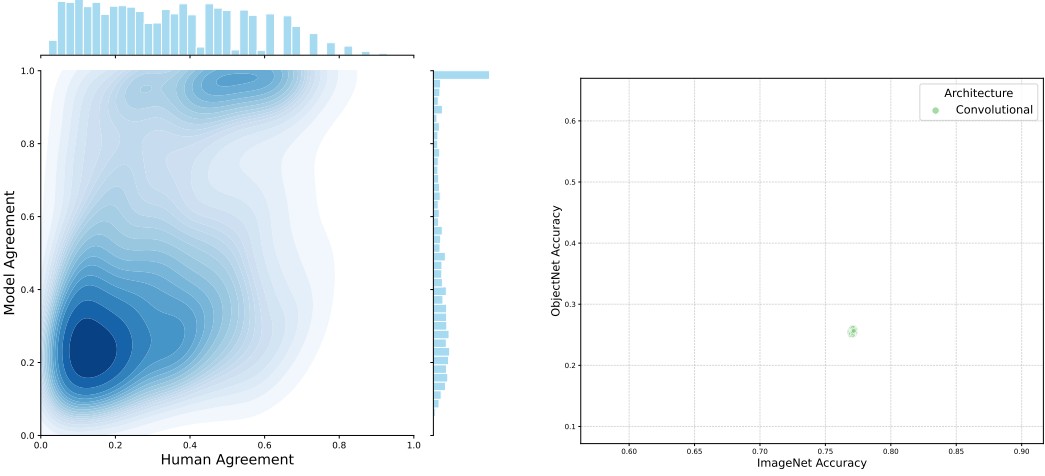

(a) Per image agreement between a population of 102 ResNet-50 models and humans.

(b) Scatter plot of ObjectNet ($y$-axis) versus ImageNet ($x$-axis) accuracy.

**Figure 5: Less diverse model population recovers the similar results** A population of 102 ResNet-50 models trained on Imagenet with the only difference being the random seed used in initialization is evaluated.

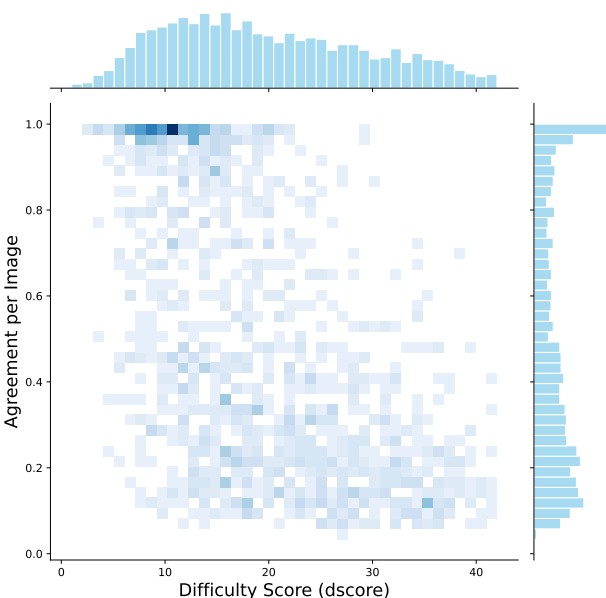

**Figure 6: Difficulty score and model agreement for ObjectNet.** Two-dimensional histogram illustrating the relationship between image difficulty for human participants, as measured by the difficulty score described in Section (2.4), and model agreement on the ObjectNet dataset. The $x$-axis represents the difficulty score for humans, with higher scores indicating more challenging images. The $y$-axis shows the level of agreement among AI vision systems, where higher values signify stronger consensus, and is analogous to the $y$-axis in Fig. (1). The color intensity represents the density of images in each bin, with darker colors indicating a higher concentration of images.

relationship, displaying a clear downward trend in model agreement as the difficulty score increases. The scatter plot demonstrates that as images become more challenging for humans (higher difficulty scores), there is a corresponding decrease in agreement among AI vision systems. This visual representation provides strong evidence for the alignment between human perceptual difficulties and AI model disagreement, highlighting that both artificial and biological visual systems struggle with similar types of challenging images.

Human agreement refers to the consistency of responses among human participants for each image as calculated using Eq. (2). Higher values indicate that most humans agree on what the image depicts. The positive correlation means that as human agreement increases, model agreement among AI vision systems also increases. This suggests that when humans consistently agree on an image's content, AI vision systems are more likely to agree as well. Conversely, images that humans find ambiguous or contentious lead to more disagreement among AI vision systems.

### 3.3 COUNTERFACTUAL LOW DIVERSITY MODEL POPULATION

To investigate the causal factors of variability within the model population leading to disagreement, we train a population of 102 ResNet-50 models He et al. (2016) over ImageNet with the only difference between individual models being the random initialization seed. In contrast to the 1032 model population collected, this population has no architecture variation or negligible performance variation (Section (3)). The reproduction of the results shown in Fig. (1) by this low diversity population as shown in Section (3) indicates the that architecture variability and training data/task variability are not significant causal factors for population disagreement.

### 3.4 ANALYSIS OF LOW-AGREEMENT IMAGES

To investigate the probable causes of image difficulty and thus disagreement, we use additional annotations of the data collected by (Mayo et al., 2023) and (Idrissi et al., 2022).

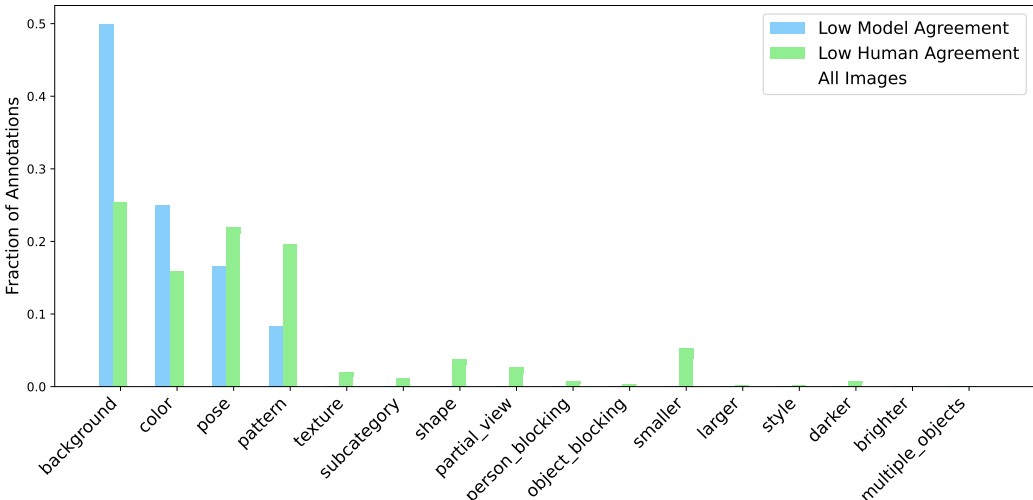

**Figure 7: Human labeled factors associated with disagreement.** Bar chart of the relative proportion of *top factors* among images from ImageNet-X that belong to low model agreement, low human agreement and all image sets. Proportions are normalized within agreement categories (*i.e.*, the blue bars sum to 1, and similarly for the green and patterned bars). The threshold for low agreement was set to 0.2 for both model and human populations, meaning that images with agreement levels below this threshold were considered "low agreement." Models deviate from humans substantially in "background" and "pattern" variations.

### 3.4.1 USING HUMAN VIEWING TIME

The minimum viewing time data represents the minimum time necessary for a majority humans to correctly classify an image. As shown in Fig. (4), disagreeable images take more time to recognize among both the model and human populations.

### 3.4.2 USING HUMAN ANNOTATIONS

We make use of the annotations provided by the ImageNet-X dataset introduced in Section (2.4). Fig. (7) displays the distribution of human-annotated visual attributes in ImageNet-X for three categories of images: those with low model agreement, those with low human agreement, and all images in the ImageNet-X dataset that have agreement scores for both models and humans.

There are notable differences between AI vision systems and humans. The large gap in proportion for "background" and "pattern" between models and humans suggests that AI systems might be more sensitive to background variations than humans and human population are more likely to disagree when pattern variations are present than models.

### 3.5 MODEL AGREEMENT PATTERNS ACROSS ACCURACY LEVELS

Lastly, we investigate how Fleiss' $\kappa$ is affected by the accuracy of model subpopulations in our sample. Fig. (8) compares Fleiss' $\kappa$ scores across ImageNet and ObjectNet datasets as a function of the mean accuracy of a subset of models from the overall model population. High agreement images (the top 1000 images in agreement score) appear to exhibit perfect agreement, indicating that the agreement level among models does not change over this set of images, presumably because the images are easy and remain easy, continuing to be correctly classified as overall population accuracy improves.

While overall Fleiss' $\kappa$ grows linearly over all images, images with low agreement (the bottom 1000 images in agreement score) among the whole population of models lack improvement in agreement until the highest accuracy level. This indicates that the models are not making progress on these examples until a certain level of competency over the full dataset is achieved. The acceleration of

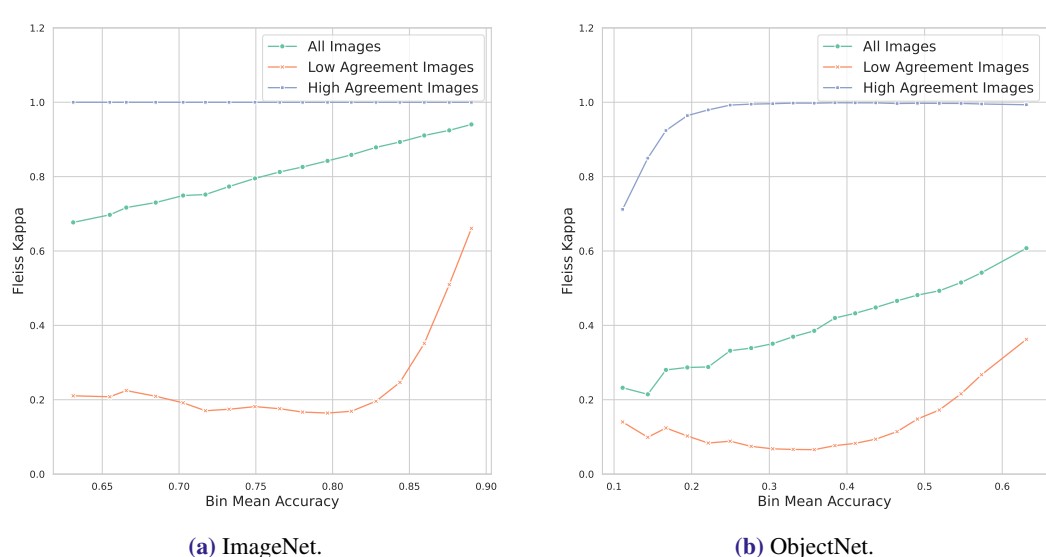

**(a)** ImageNet.

**(b)** ObjectNet.

**Figure 8: Comparison of Fleiss' $\kappa$ scores across ImageNet and ObjectNet datasets.** We plot model agreement patterns for model populations at different accuracy levels ($x$-axis) for (a) ImageNet and (b) ObjectNet. Images over which Fleiss' $\kappa$ is evaluated over are partitioned into low-agreement images and high-agreement images (*cf.* all images) according to Eq. (2) evaluated across all models. Images at low agreement levels are producing significantly lower Fleiss' $\kappa$ than high agreement and all images, even for models at high performance levels (tending to the right). This deviation is more pronounced for (b) ObjectNet.

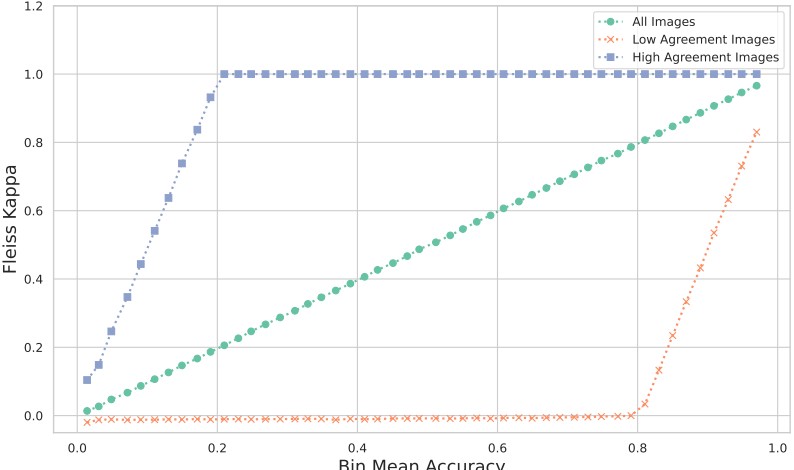

**Figure 9: Simulated reproduction of Fig. (8).** Fleiss' $\kappa$ scores for a simulated population of models with different accuracy levels. Our simulation makes the key assumption that once an image is correctly recognized, it will always be correctly recognized by more accurate models.

agreement of the low-agreement images also indicates that performant models are reaching consensus over these images faster than most images in the dataset.

We test these hypotheses for low-agreement and high-agreement images using a simulated population of models based on the assumption that any correctly classified image will always be correctly classified by a more competent (accurate) population. More details of this simulation can be found in Appendix (A.3). Fig. (9) demonstrates that our simulation does agree with the empirical lines seen in both the low-agreement images and high-agreement images in Fig. (8).

## 4 RELATED WORK

**Error analysis.**   Geirhos et al. (2020) introduced the concept of error consistency to vision modeling in order to better quantify the behavioral alignment of artificial vision systems to human vision. Complementary to accuracy, their method quantifies whether two decision-making systems make the *same* errors on the same visual stimuli, which provides further insights into the similarity of processing strategies between different models or between models and humans. Rather than concentrating on error consistency between two systems, we turn our focus towards *population*-level consistency by analyzing the inputs that generate the most divergent responses across a diverse population of AI vision systems. This approach allows us to explore the nature of model disagreement and its relationship to human perception, even in cases in which individual models may not match the errors of individual humans. Like a complete treatment of accuracy and error-consistency, our analysis does not require any knowledge of the true label or the existence of one for a given stimuli, making it suitable for unannotated data which may be more readily available.

**Synthetic stimuli.**   In contrast to the approach of Golan et al. (2020), which synthesizes "controversial" stimuli through an optimization procedure involving backpropagation through a vision system, our work focuses on analyzing disagreement between models on *naturalistic* samples that either already exist or are generated without optimization. This approach allows us to assess models' inductive biases and generalization abilities under more realistic conditions, providing insights into how the models might behave when deployed in the wild, where they may encounter novel stimuli that are not purposefully crafted to expose their differences. Furthermore, many artificial perturbations to stimuli impact the original labeling and features, which can no longer be used without making assumptions about the nature of the perturbation (Ilyas et al., 2019).

**Metamers.**   Metamers are distinct visual stimuli that are perceptually indistinguishable to a specific perceptual, often visual, system (Freeman & Simoncelli, 2011). In contrast, our study focuses on identifying single stimuli that evoke divergent responses across a diverse population of AI vision models. While metamers explore how different inputs can be perceived uniformly by a single system, our work examines how the same input can be interpreted variably by multiple systems. This fundamental difference highlights two distinct dimensions of visual representation: invariance within a system versus variability across systems.

## 5 CONCLUSION

Our study of disagreement among model and human populations in vision challenges the assumption present in prior work that disagreement among AI systems is unrelated to human visual processing. We demonstrate a correspondence between AI and human disagreement on naturalistic stimuli, revealing that image difficulty drives disagreement in both populations, and we provide evidence that the dominant factors driving disagreement in AI systems also cause disagreement in human visual perception. Our results suggest that individual differences in AI vision systems may overlap with those in human visual processing rather than being unique artificial limitations, which prescribes more population-level comparisons between AI and human vision systems rather than model-to-individual comparisons.

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

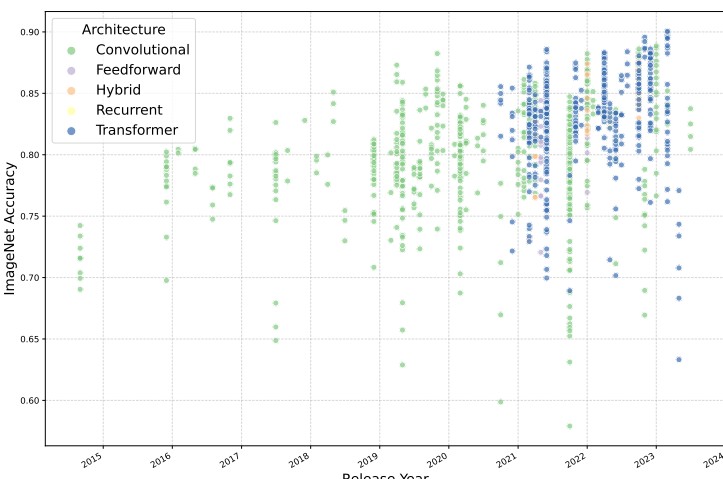

**Figure 10: Survey of the AI vision system population by release date.** Evolution of ImageNet accuracy over time (2014-2024, discretized by month) for the AI vision systems in the population described in Section (2.2). The upward trend demonstrates consistent improvement in model performance, with more recent architectures showing rapid progress in recent years.

## A APPENDIX

### A.1 AI SYSTEM INVENTORY

Below is a sample of the first 50 models by alphabetical model ID (which identifies hyperparameters like architecture and pre-training and fine-tuning datasets) of the AI system population described in Section (2.2); the complete identification of all models is given in the accompanying code repository.

**model ID**
```
bat_resnext_26_256_classification_imagenet_1k
beit_base_patch16_224_mim_imagenet_21k_ft_classification_imagenet_21k_imagenet_1k
beit_base_patch16_384_mim_imagenet_21k_ft_classification_imagenet_21k_imagenet_1k
beit_large_patch16_224_mim_imagenet_21k_ft_classification_imagenet_21k_imagenet_1k
beit_large_patch16_384_mim_imagenet_21k_ft_classification_imagenet_21k_imagenet_1k
beit_large_patch16_512_mim_imagenet_21k_ft_classification_imagenet_21k_imagenet_1k
beitv2_base_patch16_224_mim_imagenet_1k_ft_classification_imagenet_1k
beitv2_base_patch16_224_mim_imagenet_1k_ft_classification_imagenet_21k_imagenet_1k
beitv2_large_patch16_224_mim_imagenet_1k_ft_classification_imagenet_1k
beitv2_large_patch16_224_mim_imagenet_1k_ft_classification_imagenet_21k_imagenet_1k
botnet_26_256_c1_imagenet_1k
caformer_base_36_224_classification_imagenet_1k
caformer_base_36_224_classification_imagenet_21k_ft_classification_imagenet_1k
caformer_base_36_384_classification_imagenet_1k
caformer_base_36_384_classification_imagenet_21k_ft_classification_imagenet_1k
caformer_medium_36_224_classification_imagenet_1k
caformer_medium_36_224_classification_imagenet_21k_ft_classification_imagenet_1k
caformer_medium_36_384_classification_imagenet_1k
caformer_medium_36_384_classification_imagenet_21k_ft_classification_imagenet_1k
caformer_small_18_224_classification_imagenet_1k
caformer_small_18_224_classification_imagenet_21k_ft_classification_imagenet_1k
caformer_small_18_384_classification_imagenet_1k
caformer_small_18_384_classification_imagenet_21k_ft_classification_imagenet_1k
caformer_small_36_224_classification_imagenet_1k
caformer_small_36_224_classification_imagenet_21k_ft_classification_imagenet_1k
caformer_small_36_384_classification_imagenet_1k
caformer_small_36_384_classification_imagenet_21k_ft_classification_imagenet_1k
cait_medium_36_384_distillation_imagenet_1k
cait_medium_48_448_distillation_imagenet_1k
cait_small_24_224_distillation_imagenet_1k
cait_small_24_384_distillation_imagenet_1k
cait_small_36_384_distillation_imagenet_1k
cait_xsmall_24_384_distillation_imagenet_1k
cait_xxsmall_24_224_distillation_imagenet_1k
cait_xxsmall_24_384_distillation_imagenet_1k
cait_xxsmall_36_224_distillation_imagenet_1k
cait_xxsmall_36_384_distillation_imagenet_1k
coat_lite_medium_224_classification_imagenet_1k
coat_lite_medium_384_classification_imagenet_1k
coat_lite_mini_224_classification_imagenet_1k
coat_lite_small_224_classification_imagenet_1k
coat_lite_tiny_224_classification_imagenet_1k
coat_mini_224_classification_imagenet_1k
coat_small_224_classification_imagenet_1k
coat_tiny_224_classification_imagenet_1k
coatnet_224_sw_imagenet_12k_ft_classification_imagenet_1k
coatnet_224_sw_imagenet_1k
coatnet_bn_224_sw_imagenet_1k
coatnet_nano_224_sw_imagenet_1k
coatnet_rmlp_224_sw_imagenet_12k_ft_classification_imagenet_1k
coatnet_rmlp_224_sw_imagenet_1k
```

## A.2 DERIVATION OF FLEISS' $\kappa$ WITH INCREASING ACCURACY

We derive how Fleiss' $\kappa$ (Eq. (4)) changes for a population of raters with increasing performance.

Let:

- $A$: Accuracy of the AI systems (probability of correct classification).

Assume:

- Each model correctly classifies a stimulus with probability $A$.
- When incorrect, a model randomly selects one of the $(k-1)$ incorrect categories with equal probability.

For a given stimulus $i$, the expected number of models assigning category $j$ is:

- **Correct category:** $T_{i,\text{correct}} = M \times A$
- **Incorrect category (for each incorrect category $j$):** $T_{ij} = M \times \frac{1-A}{k-1}$

Using Eq. (2):

$$p_{\text{agree},i} = \frac{\sum_{j=1}^{k} T_{ij}(T_{ij} - 1)}{M(M - 1)} \tag{5}$$

Substituting $T_{i,\text{correct}}$ and $T_{ij}$:

$$p_{\text{agree},i} = \frac{T_{i,\text{correct}}(T_{i,\text{correct}} - 1) + (k - 1) \times T_{ij}(T_{ij} - 1)}{M(M - 1)} \tag{6}$$

$$= \frac{[MA(MA - 1)] + (k - 1)\left[M\frac{1-A}{k-1}\left(M\frac{1-A}{k-1} - 1\right)\right]}{M(M - 1)} \tag{7}$$

Correct category term:

$$MA(MA - 1) = M^2 A^2 - MA \tag{8}$$

Incorrect categories term:

$$(k - 1)\left[M\frac{1 - A}{k - 1}\right]^2 - M\frac{1 - A}{k - 1} = \frac{M^2(1 - A)^2}{k - 1} - M(1 - A) \tag{9}$$

Total numerator:

$$(M^2 A^2 - MA) + \left(\frac{M^2(1 - A)^2}{k - 1} - M(1 - A)\right) \tag{10}$$

Simplify combined numerator:

$$M^2 A^2 + \frac{M^2(1 - A)^2}{k - 1} - M(A + 1 - A) = M^2 A^2 + \frac{M^2(1 - A)^2}{k - 1} - M \tag{11}$$

Denominator:

$$M(M - 1) = M^2 - M \tag{12}$$

Per-stimulus agreement:

$$p_{\text{agree},i} = \frac{M^2 A^2 + \frac{M^2(1-A)^2}{k-1} - M}{M^2 - M} \tag{13}$$

Divide numerator and denominator by $m^2$:

$$p_{\text{agree},i} = \frac{A^2 + \frac{(1-A)^2}{k-1} - \frac{1}{M}}{1 - \frac{1}{M}} \tag{14}$$

We can use this formulation of agreement per image and substitute this term into Eq. (2) and Eq. (4) to derive a formulation for Fleiss' $\kappa$ in terms of accuracy under the provided assumptions.

For large $M$ we can approximate this to:

$$p_{\text{agree},i} \approx A^2 + \frac{(1-A)^2}{k-1} \ . \tag{15}$$

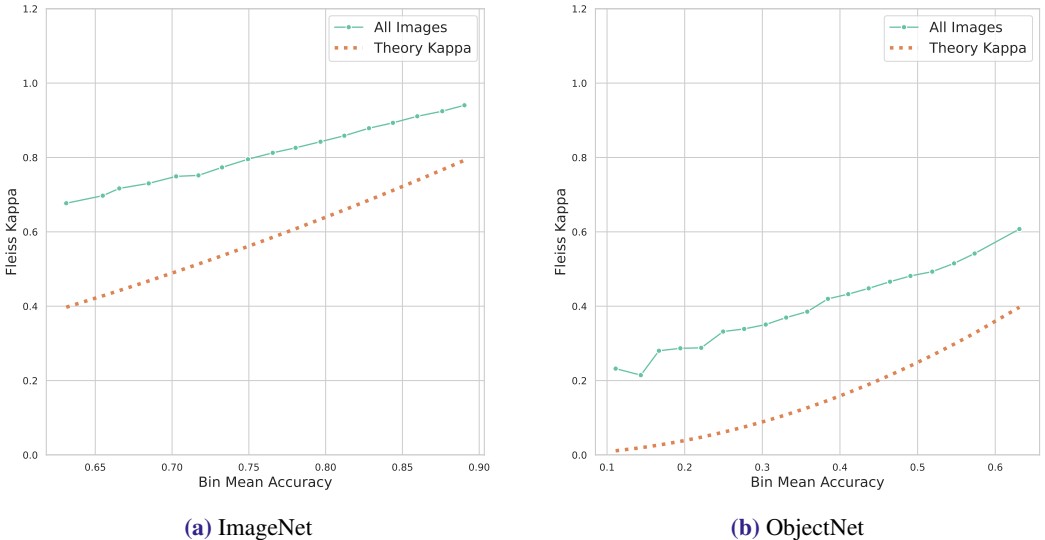

(a) ImageNet

(b) ObjectNet

**Figure 11:** Comparison of empirical Fleiss' $\kappa$ to the Fleiss' derived using Eq. (14)

The deviations of the empirical Fleiss' $\kappa$ from the derivation shown in Fig. (11) show that our models do not follow the assumptions implied in the derivation. In particular, the consistent larger agreement indicates that models are not agreeing in random uniform way across images in ImageNet and ObjectNet.

### A.3 FLEISS' $\kappa$ WITH WITH A SIMULATED POPULATION OF MODELS

We constructed a simulation of Fleiss' $\kappa$ among our population of models adding one additional assumption: Image classifications are learned strictly in order. In particular, a model with 80% classification accuracy will correctly classify all of the images that a 40% accuracy model gets right plus and additional 40% of images in the set of images evaluated. We implemented this simulation by simply giving $N$ images a random unique learnability ranking from 1 to $N$. A simulated model then correctly classifies the first $C$ images where $C/N$ matches the models average accuracy. The models predictions for the remaining $N - C$ predictions are random.

Running this simulation with the parameters $N = 50\,000$, $M = 1032$, $k = 50$, sampling average model accuracy randomly between 0 and 100, results in Fig. (9).

---

**Algorithm 1** Generation of model population with ordered learnability.

---

1: **Initialize parameters**
2: $N \leftarrow 50000$      // Number of images
3: $M \leftarrow 1032$      // Number of models
4: $k \leftarrow 50$      // Number of classes
5:
6: **Assign ground truth labels to images**
7: $images\_per\_class \leftarrow \lfloor N/k \rfloor$
8: $remainder \leftarrow N \bmod k$
9: $GT\_labels \leftarrow$ empty list
10: **for** $class \leftarrow 1$ to $k$ **do**
11:      **if** $class \leq remainder$ **then**
12:          $n \leftarrow images\_per\_class + 1$
13:      **else**
14:          $n \leftarrow images\_per\_class$
15:      **end if**
16:      Append $n$ instances of $class$ to $GT\_labels$
17: **end for**
18: Randomly shuffle $GT\_labels$
19:
20: **Assign unique learnability rankings to images**
21: $image\_indices \leftarrow$ Random permutation of $\{1, 2, \ldots, N\}$
22:
23: **Assign random accuracies to models**
24: **for** $m \leftarrow 1$ to $M$ **do**
25:      $accuracy[m] \leftarrow$ Random value in $[0.65, 0.90]$
26: **end for**
27:
28: **Generate predictions for each model**
29: **for** $m \leftarrow 1$ to $M$ **do**
30:      $C \leftarrow$ Round $(accuracy[m] \times N)$
31:      $correct\_indices \leftarrow image\_indices[1 : C]$
32:      $incorrect\_indices \leftarrow image\_indices[C + 1 : N]$
33:      **for** each $i$ in $correct\_indices$ **do**
34:          $predictions[m, i] \leftarrow GT\_labels[i]$
35:      **end for**
36:      **for** each $i$ in $incorrect\_indices$ **do**
37:          $incorrect\_classes \leftarrow \{1, 2, \ldots, k\} \setminus \{GT\_labels[i]\}$
38:          $predictions[m, i] \leftarrow$ Random choice from $incorrect\_classes$
39:      **end for**
40: **end for**

---

### A.4    HUMAN AGREEMENT VS. MODEL AGREEMENT

Fig. (12) compares the density plot of models versus human agreement for ImageNet and ObjectNet. In ObjectNet, which contains more challenging natural images, AI models show stronger alignment with human judgments on images that cause disagreement rather than agreement. This trend is less pronounced in ImageNet. The dominant agreement in ImageNet is likely due to the model population being trained or fine-tuned on ImageNet making this task easier than ObjectNet.

### A.5    ADDITIONAL IMAGES SELECTED BASED ON AGREEMENT SCORE

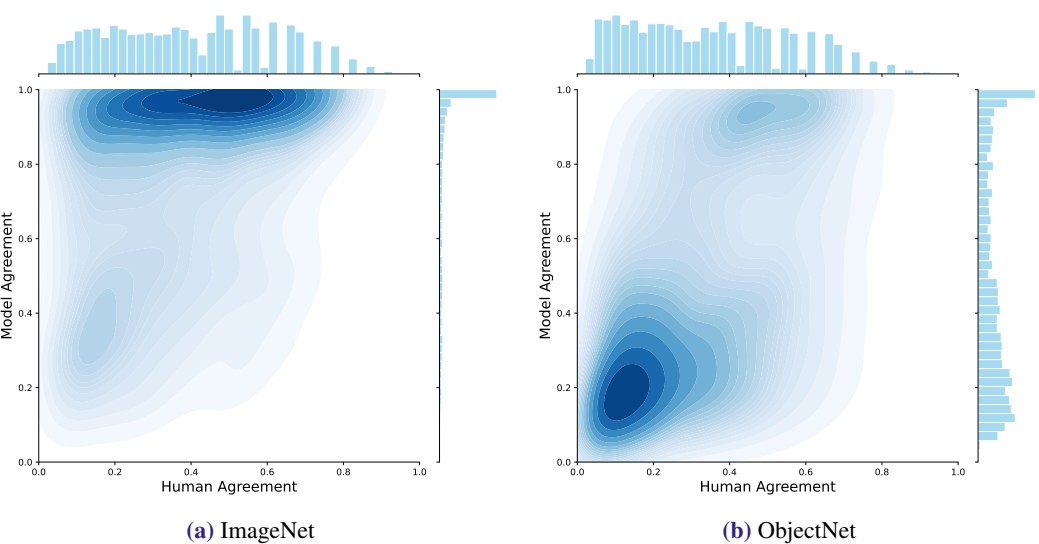

(a) ImageNet  (b) ObjectNet

**Figure 12:** Comparison of agreement levels between AI vision models and humans for (a) ImageNet and (b) ObjectNet datasets. Each point represents an image, with its position indicating the agreement level among 1032 AI vision systems ($y$-axis) and 42 human participants ($x$-axis).

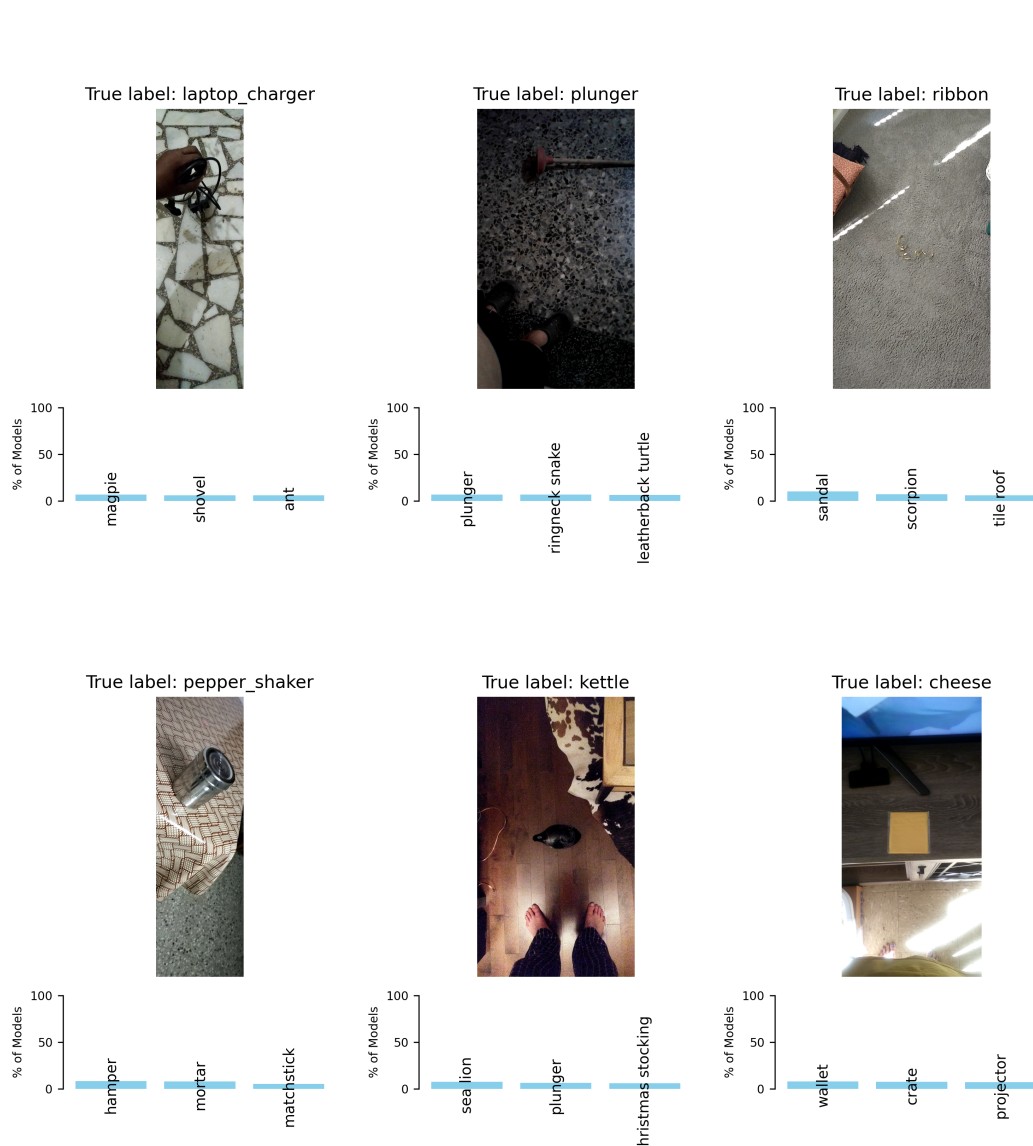

**Figure 13:** Lowest agreement images from ObjectNet

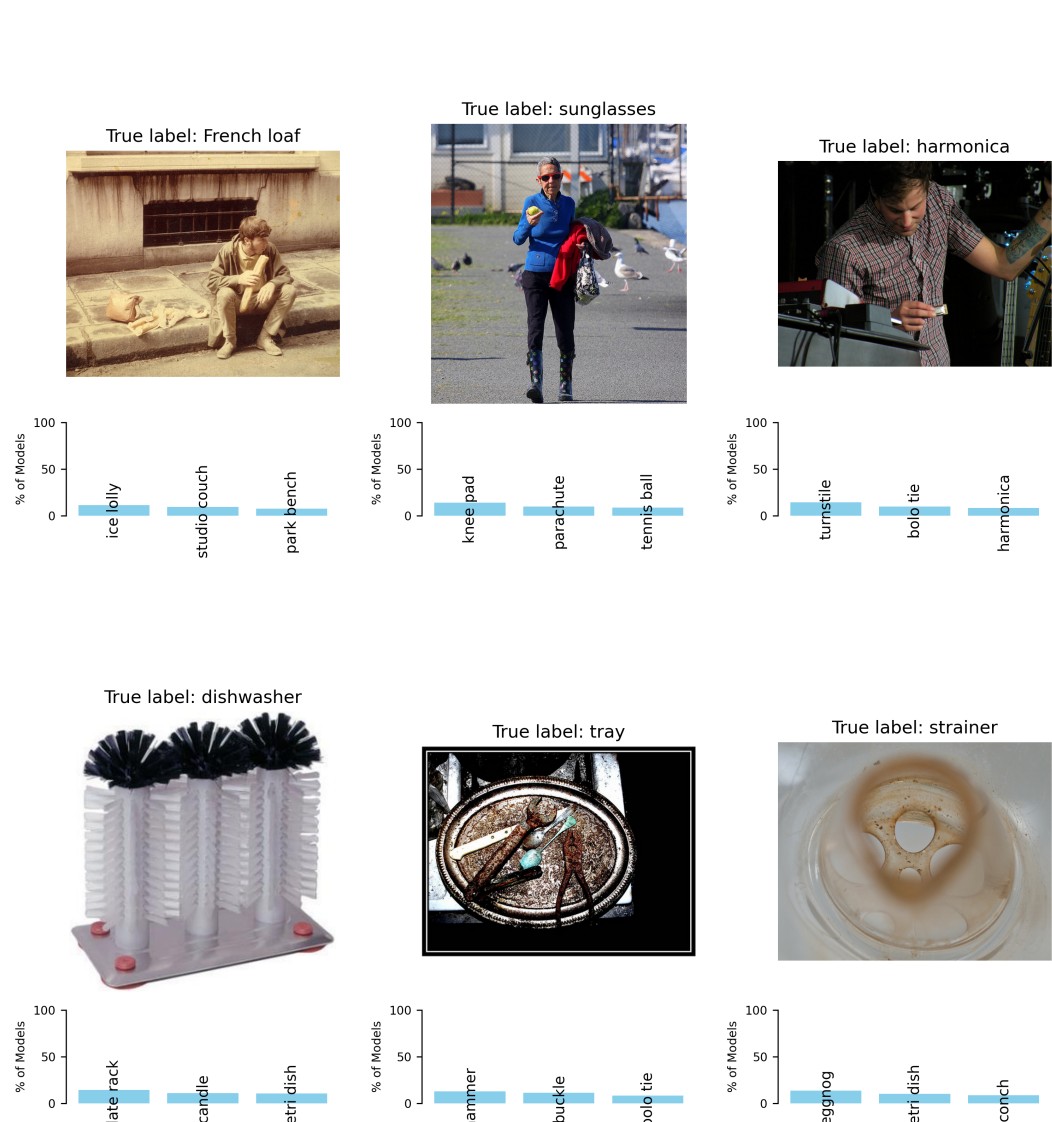

**Figure 14:** Lowest agreement images from ImageNet

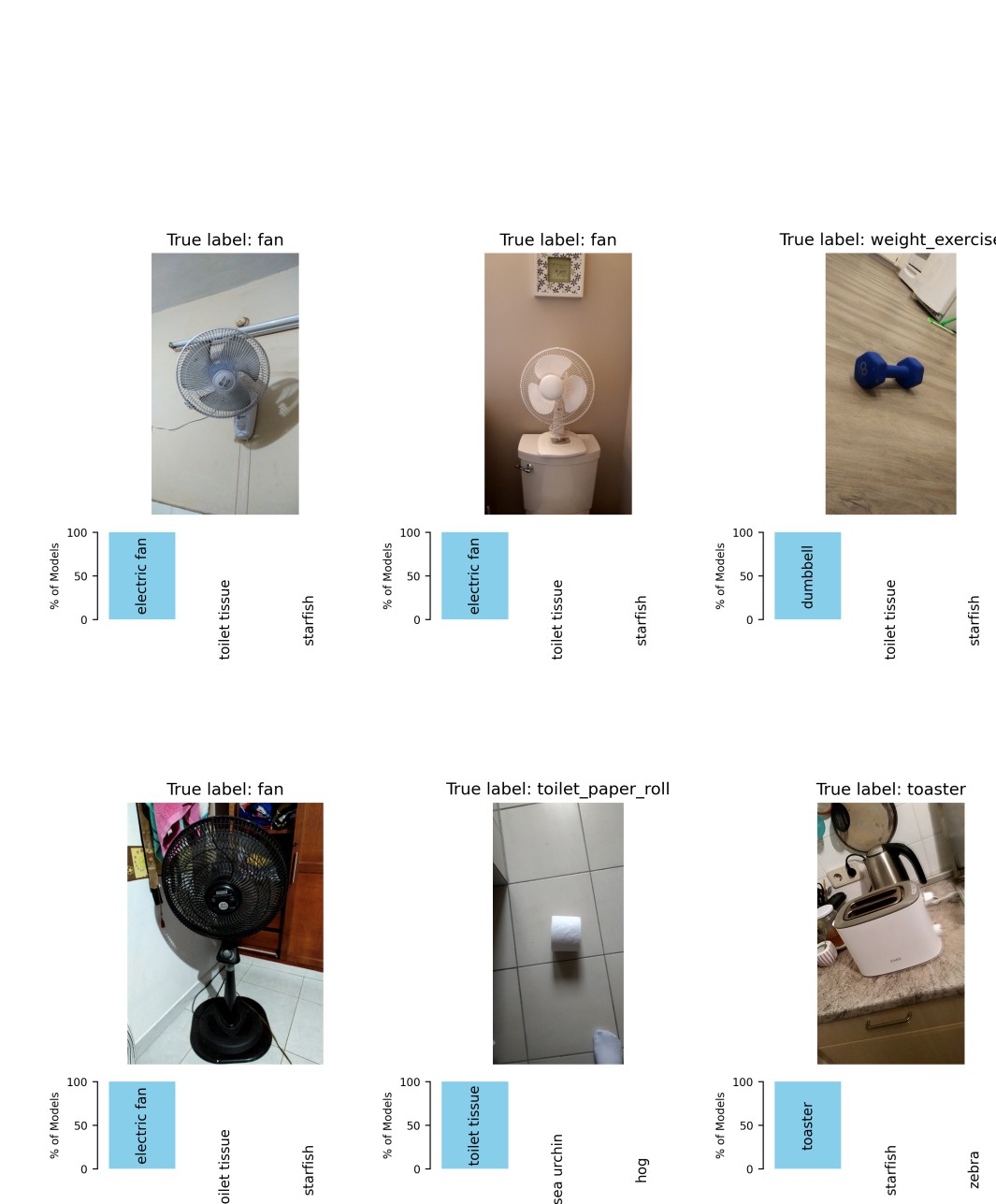

**Figure 15:** Highest agreement images from ObjectNet

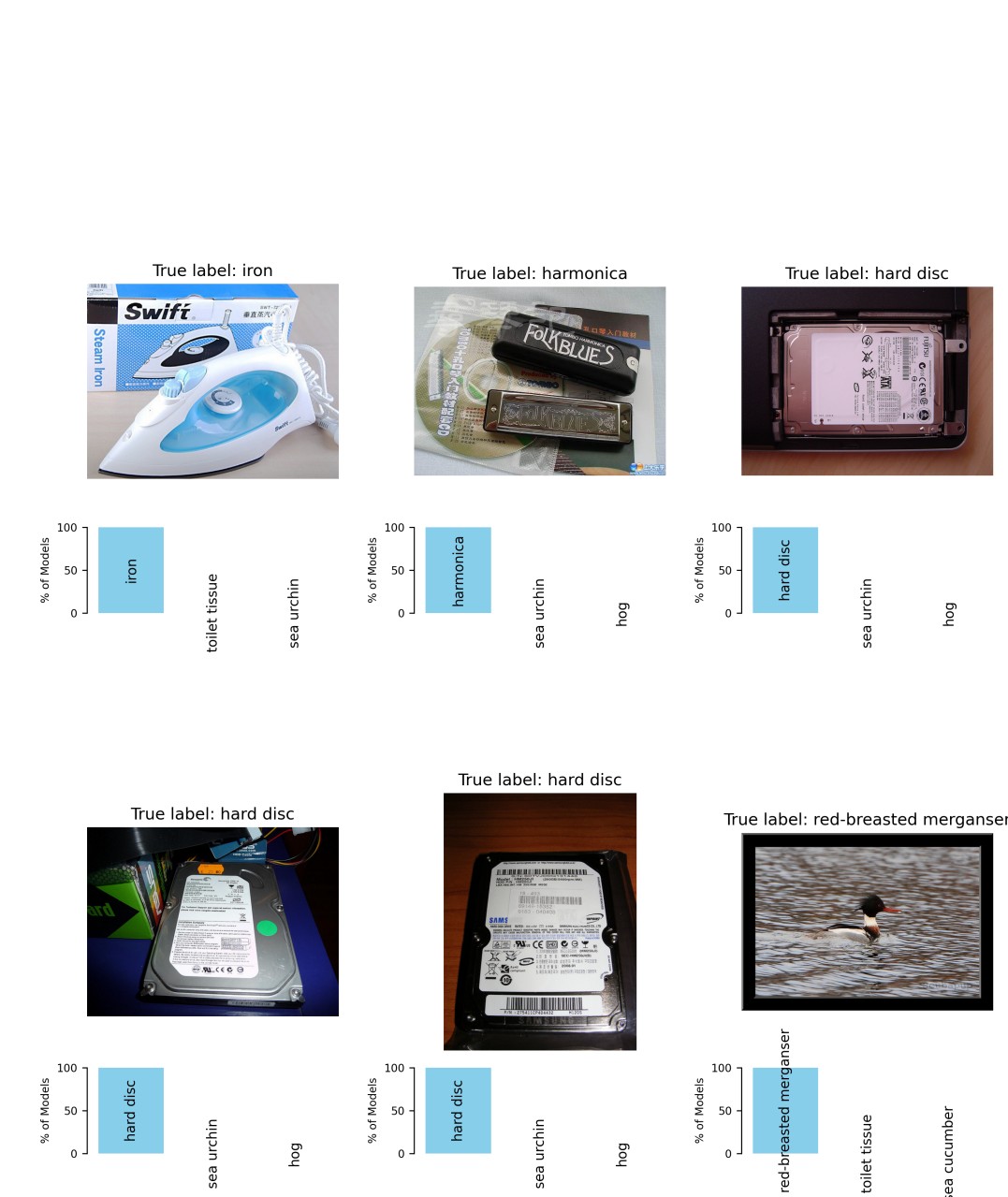

**Figure 16:** Highest Agreement Images from ImageNet

