# OpenReview forum: "Let’s disagree to agree: Evaluating collective disagreement among AI vision systems"
_ICLR.cc/2025/Conference — Submitted to ICLR 2025_

### Official Review · Reviewer_XaeV · 2024-10-28

**Soundness:** 2
**Presentation:** 2
**Contribution:** 3
**Rating:** 3
**Confidence:** 3

**Summary:**

The article explores the disagreement behaviors of AI vision systems, diverging from traditional approaches that compare individual AI models to biological vision. Instead, this study investigates patterns of agreement and disagreement among a diverse population of AI models by measuring "aggregate disagreement" across model outputs. It aims to determine which inputs produce the most divergent responses among models and assesses whether these inputs also create discrepancies between AI systems and human perception.
A significant finding is that even images causing high disagreement among AI models often align with human perceptual challenges. This alignment suggests that the limitations in AI models mirror similar perceptual difficulties in humans, offering valuable insights into AI-human vision comparisons at a population level. This work contributes to the field by reframing disagreement not as an intrinsic limitation of AI systems but as an opportunity to study the shared perceptual challenges between artificial and human vision systems.

**Strengths:**

1.Innovative Research Topic:
The authors investigate an intriguing and novel research area by examining AI model and human visual disagreements at a population level. This approach is unique in that it moves beyond individual model comparisons to analyze the collective behavior of AI vision systems.
2.New Method for Measuring Human-AI Discrepancy:
By introducing a method to measure disagreement at the population level, the study provides a new way to quantify the difference between AI models and human perception, adding a meaningful metric to the field.
3.Focus on Naturalistic Stimuli:
Unlike prior work that often uses synthetic stimuli, this study investigates the properties of naturalistic stimuli that elicit the most disagreement among AI models, making its findings more applicable to real-world scenarios.
4.Insights into AI-Human Perceptual Alignment:
The article provides evidence suggesting that disagreements among AI systems are influenced by aspects of human visual perception, particularly in image difficulty, as measured by human behavioral data. This insight supports the idea that individual differences in AI vision systems may reflect differences in human visual processing rather than inherent AI limitations.

**Weaknesses:**

1.Limited Analysis of Outlier Cases:
The authors report correlations between model agreement and human behavioral measures, but they do not analyze specific cases where model agreement is high but human difficulty is low, or vice versa. Such an analysis could provide deeper insights into unique points of divergence.
2.Lack of Architecture-Specific Insights:
Although multiple model architectures are included in the study, the authors do not analyze how different architectures impact the results. This oversight limits the understanding of how architectural variations might contribute to AI-human agreement or disagreement on challenging stimuli.
3.No Exploration of Methods to Reduce Disagreement:
While the study highlights greater disagreement on images of higher human difficulty, it does not explore whether certain methods, such as targeted model adjustments or expanded training datasets, could reduce this disagreement and improve alignment with human perception.
4.Insufficient Citations of Related Work on AI-Human Disagreement:
Prior research has shown that there are links between AI-human disagreement and human visual processing at the individual model level, yet the authors do not reference these foundational works. Including these citations could strengthen their arguments by situating the study within the existing body of research.

**Questions:**

1.Did the authors consider analyzing cases where model agreement is high but human difficulty is low, or where model agreement is low but human difficulty is high? Such cases might offer valuable insights into the nuanced differences between AI model behavior and human perception.
2.Although multiple architectures were included, why did the authors not explore the impact of different architectures on the experimental results?
3.Can the higher disagreement on challenging human images be reduced through specific adjustments to models or training datasets?
4.Previous research has shown links between AI-human disagreement and human visual processing at the individual model level. Why were these relevant studies not carefully discussed in the related work section?

If the authors can address these issues, I would be happy to raise my score.

---

### Official Review · Reviewer_tvpj · 2024-11-01

**Soundness:** 3
**Presentation:** 3
**Contribution:** 2
**Rating:** 5
**Confidence:** 3

**Summary:**

This paper investigates correlations between populations of humans and object-recognition systems on object-classification disagreements.  The results show that there is significant correlation between human and model population disagreements, as well as between human minimum viewing time and model disagreements.  The results support the hypothesis that this correlation is driven by aspects of human visual perception that makes certain aspects of images difficult to classify.

**Strengths:**

The experiments seem solid and the results are well-presented.  The authors tested over 1,000 different models, including CNNs, ViTs, and hybrid models.  The paper goes more deeply than just giving correlation statistics, and investigates what features low-agreement images have in common.

**Weaknesses:**

I'm not sure how useful these results are, either for understanding human or machine vision, or for improving machine vision systems.  A useful result would point in a new direction for experiments (to better understand underlying mechanisms) and/or architectural improvements.  But what are the next steps with these results?  The authors did not address this or make the case that these results are important for the field.

The paper states: "In this work, we challenge the assumption that disagreement among AI systems is intrinsic to these systems and unrelated to aspects of human visual processing".  But what are the citations for this assumption?

I didn't understand, in second paragraph, how this assumption "aligns with standard approachs for comparing internal representations of AI and biological vision, such as representational similarity analysis" or how it is "explicit in behavioral extrapolation tests" -- this needs better explanation.

**Questions:**

The paper states: "AI systems might be more sensitive to background variations than humans and human population are more likely to disagree when pattern variations are present".  Explain what "pattern" refers to here.

When giving models' accuracy on ImageNet and ObjectNet datasets, are you using top-5 or top-1 accuracy?  What about for humans?

Figure 7: What is "Bin Mean Accuracy"?

---

### Official Review · Reviewer_ihcH · 2024-11-03

**Soundness:** 3
**Presentation:** 2
**Contribution:** 3
**Rating:** 6
**Confidence:** 3

**Summary:**

The paper brings a new point from population-level comparisons between AI and human vision systems, different from the previous individual Ai and human comparison. The authors conduct experiments using a large population of 1032 models and a previous user study with 42 human participants. They use Fleiss' kappa to quantify the level of agreement and find out a few interesting points on the correlation between AI model (dis)agreement and human (dis)agreement. They claim that the low agreement on hard images is due to intrinsic perceptual challenges shared by both AI and humans instead of model structure limitations.

**Strengths:**

The strengths:

- brings a novel view from population-level comparison of AI and human on vision systems.

- conduct extensive experiments on a large population AI models

- Interesting findings on AI models not perform well on difficult images due to perceptual challenges that human faces as well.

**Weaknesses:**

Weaknesses of this paper include:

- Some findings are quite intuitive, for example, the correlation between AI (dis)agreement and human (dis)agreement. This probably is due to the labels are created by humans.

- 42 participants from user study might be a bit bias. May conduct a few more user studies and combine with previous data.

- The image style does not look very good, some images are taking too many spaces but contain relatively few contents.

- at line 402, "Images at low agreement levels are produce...", should be "... are producing..."

**Questions:**

- In Fig 1, it is a bit surprising that there are very few images with high human agreement from the top histogram, which means humans rarely have full agreement on images. Could you explain possible reasons behind this?

- If humans and AI cannot recognize the difficult images or the edge-case images, it means vision alone cannot solve the problem and we probably do not have a better solution using only vision. What other benefits could it bring to us if we study more on the difficult images? In other words, how does studying the edge-case images help?

---

> ### Author Response · Authors · 2024-11-18
>
> We appreciate the clear and insightful review. We would like to address some comments:
>
> > Some findings are quite intuitive, for example, the correlation between AI (dis)agreement and human (dis)agreement. This probably is due to the labels are created by humans.
>
> We agree that there is a an intuitive aspect that the models are trained on labels created by humans. But we think the intuition is less obvious than at first appearance because all models were trained on the same singular label generated by humans. We see that despite all models being trained to provide the same labeling over the training set, they disagree on held out images in a way that is similar to human population would disagree. Despite no labeling variation during training, the prediction variation acquired by the population of models (as measured through disagreement) is similar to a human population. We will update the draft to make this point more clear.
>
> > 42 participants from user study might be a bit bias. May conduct a few more user studies and combine with previous data.
>
> Sorry for the confusion. There are actually 2,647 human participants from the user study we used. We've updated the draft to make this more clear. There are 42 predictions per image (7 different subjects seeing each image at one of six timings). To eliminate memory recall effects, each subject needs to be viewing the image for the first time.
>
> > The image style does not look very good, some images are taking too many spaces but contain relatively few contents.
>
> Thank you for the suggestion. We have adjusted the figures to be more space efficient by making better use of the aspect ratio with more information presented in each row.
>
> > In Fig 1, it is a bit surprising that there are very few images with high human agreement from the top histogram, which means humans rarely have full agreement on images. Could you explain possible reasons behind this?
>
> ObjectNet was collected to be a challenging dataset to models that perform well on ImageNet. Given the large range of viewing times (from 17ms to 10s), humans also have difficulty in 100% correctly predicting the images correctly across all viewing times. Interestingly, it is actually the the shorter viewing times where humans tend to agree more than longer ones.
>
> > If humans and AI cannot recognize the difficult images or the edge-case images, it means vision alone cannot solve the problem and we probably do not have a better solution using only vision. What other benefits could it bring to us if we study more on the difficult images? In other words, how does studying the edge-case images help?
>
> We view edge cases something akin to optical illusions for visual intelligence. Much like the famous blue dress image (https://en.wikipedia.org/wiki/The_dress), edge cases that lead to disagreement could reveal compelling discrepancies or similarities between populations of visual intelligences.

---

### Official Review · Reviewer_pHQG · 2024-11-03

**Soundness:** 2
**Presentation:** 3
**Contribution:** 2
**Rating:** 3
**Confidence:** 3

**Summary:**

This paper attempts to establish similarity between artificial and biological vision by showing that populations of AI models and populations of humans show intra-group disagreement on the same stimuli. It motivates itself by claiming that prior work shows disagreement among models being a function of limitations in their development, rather than expressions of an underlying mechanism in both AI and human vision.

The paper defines agreement as Fleiss' $\kappa$ for an image, calculated over a population of vision systems. It surveys ~40 humans and ~1000 models, trying CNNs, ViTs, and hybrids and varying model size, dataset size, and training methods (pretraining and finetuning). It also uses human minimum viewing time and difficulty score as comparison metrics.

Results show:
- All metrics appear to correlate with model agreement in intuitive ways - not strong correlations, but significant and all in the intuitive direction
- The clearest relationship is for low-difficulty high-model agreement images
The paper takes human-annotated visual attributes from the ImageNet-X dataset, in which humans annotated what aspects of an image make it difficult to classify. The paper showed that for both low-human agreement and low-model agreement images, the percent of images with each top difficulty factor shows similar relative influence - the percentage of images for each factor decreases in mostly the same order for both humans and models. The most influential factors are found to be background, pose, color, pattern, and "smaller".

The paper also shows that model agreement increases as accuracy increases.

The paper then positions itself against other error analysis-related works, works that use synthetic stimuli to assess differences, and metamers (this being an opposite of a metamer).

**Strengths:**

### Quality
- Good problem setup: well-defined, statistical choices make sense, and experiences overall make sense (I will list a couple exceptions in the weaknesses)
- Good application of ImageNet-X to get systematic error analysis on naturalistic images
- Comparing to a population of models seems promising

### Clarity
- Writing style is very clear. I rarely felt confused when reading the paper, and the structure made sense.
- Figures are well-designed. They are the most useful aspect for building intuition about the results - they look good, and show the right concepts.
- Explanation of Fleiss' $\kappa$ helps build intuition for what "agreement" means, and also helps strengthen the experimental design choices

**Weaknesses:**

### Quality
#### Problem
- Motivation isn't that convincing - the paper claims that the typical assumption around model errors is "intrinsic to these systems and unrelated to aspects of human visual processing." But that isn't always the case - I think ambiguous images (which seem to be the crux of this paper) are not only known to be difficult for models just as they are difficult for humans, but are easily cited by most researchers as a cause of model error and likely disagreement
  - The paper also claims evidence that "disagreement among AI vision systems is driven by aspects of human visual perception, particularly image difficulty" - it's worth nothing that classifications are a human concept, not an inherent property of the image, and training data reflects that. Maybe the paper isn't directly making this claim, but it seems that it's suggesting there are similar mechanisms between models (at least model populations) and humans that drive disagreement; I'd argue that these images are simply actually ambiguous, the classification is a product of human reaction to ambiguity, the training data is also a product of human reaction to ambiguity, and the model directly encodes that rather than showing an interesting emergent behavior.
- Data on variations of models is limited to a list in the appendix - would be good to be given a structured representation of the variations in a table

#### Results
- Though the correlation coefficients are nontrivial and the figures line up with them, and I wouldn't expect strong correlations for such a high-dimensional problem, the figures do show a lot of spread.
- This also make the results seem less surprising - from both this and figure 6, where we see the factors being "background","pose", "color", "pattern", and "smaller", it seems that the difficult images are simply truly ambiguous. It's not a matter of ML fallibility, but I wouldn't expect it to be. It's also not an underlying surprising mechanism in human vision that makes humans fallible on them. The images are ambiguous and the humans who labeled them probably weren't completely sure what to label them. Even if we call it a shared mechanism/underlying principle of human vision, it's not surprising or unknown.
- It makes sense that agreement increases as overall accuracy increases, but this is really not surprising. It could be that there are cases where models all classify the image as the same wrong class, but just given how training works, it's likely the original image is misclassified (or the original assumption is true). In either case, this doesn't offer an alternative to an explanation to the original assumption.

### Clarity
- Would help to have an explanation of why Fleiss' $\kappa$ is a good measure of agreement, really just intuition on how it works.
- Sections 3.1 and 3.2 don't need to be there - they explain concepts that are immediately clear from the figures.
- More descriptive statistics on the figures would help understand how predictive the results are.

### Originality and significance
- I haven't seen this framing of this problem. However, the concept itself - that ambiguous images are difficult for both humans and models - doesn't seem novel. It also doesn't seem to warrant this much formalization.

**Questions:**

- I am curious how these experiments would fare for top-5 classification - possibly for humans, not just models
- In figure 6, how should we factor in the difference in proportions between models and humans, even if the order of proportions is mostly the same? I realize you're not making this claim, but if we want to establish similar underlying mechanisms, we'd need to deal with the differences in proportion for each factor. What might this imply for future studies?
- "Images at low agreement levels are produce significantly lower Fleiss' $\kappa$ than high agreement and all images, even for models at high performance levels" - I thought that agreement is *defined* as Fleiss' $\kappa$. Am I misinterpreting? Is the point that even when models are split and Fleiss' $\kappa$ is recalculated, it is low for the images that had low Fleiss' $\kappa$ across all models? That would be more meaningful, though continues to point to images that are simply ambiguous.

---

> ### Author Response · Authors · 2024-11-12
> **Discussion of some key concerns**
>
> > **Motivation isn't that convincing**
>
> While the actual notion of disagreement among a population of models has not been measured before our submission, it has been an explicitly stated assumption that the mistakes that AI models make are distinct from the mistakes that humans make. For instance, [Geirhos et al. (NeurIPS 2020)](https://arxiv.org/abs/2006.16736) make the points:
>
> "The consistency between CNNs and human observers, however, is little above what can be expected by chance alone—indicating that humans and CNNs are likely implementing very different strategies."
>
> "We conclude that there is a substantial algorithmic difference between human observers and the investigated sixteen CNNs: humans and CNNs are very likely implementing different strategies."
>
> “Cohen’s $\kappa$ for CNN-human consistency is very low for both models (`.068` for ResNet-50; `066` for CORnetS) compared to `.331` for human-human consistency.”
>
> Furthermore, [Geirhos et al. (NeurIPS 2018)](https://arxiv.org/abs/1808.08750) make the point:
>
> “Additionally, we find progressively diverging patterns of classification errors between humans and DNNs with weaker signals.”
>
> > **the training data is also a product of human reaction to ambiguity**
>
> Although the labels come from humans, the labels this model population sees are unanimous amongst all models. Therefore, we see that despite all models being trained to provide the same labeling over the training set, they disagree on held out images in a way that is similar to human populations.

---

> > ### Author Response · Authors · 2024-11-12
> > **Also thank you for the review**
> >
> > We're responding to the reviews as timely as possible because the points you have made are important and we would like to discuss further.

---

> > ### Comment · Reviewer_pHQG · 2024-11-12
> > **Thanks for the comments - score not changed.**
> >
> > - The quotes from Geirhos et al. are mainly about strategies. It is fair to say that the quote "the consistency between CNNs and human observers, however, is little above what can be expected by chance alone" and fig 1 in Geirhos et al. are about mistakes, not just strategies, and it does raise questions that we observe human-model consistency that seems driven by the image/label rather than random chance. However, that means your paper demonstrates the need for context in Geirhos et al. It does not mean that the consistency you show is nonobvious or a significant contribution - that's a separate question.
> >
> > - So, let's come to that question and your second response point above. You said something similar to reviewer vUbw - "humans and machines are potentially challenged in the same way by the same images. We think this is not obvious at the population level because all the models were individually trained/fine-tuned on the same labels, so there's no ambiguity in their training, but ambiguity and disagreement nonetheless arises. And that ambiguity and disagreement appears to be aligned with populations of humans." And to me, "Although the labels come from humans, the labels this model population sees are unanimous amongst all models. Therefore, we see that despite all models being trained to provide the same labeling over the training set, they disagree on held out images in a way that is similar to human populations."
> >   - You're right to say that the models were trained on the same labels for each image and that takes away one source of ambiguity.
> >   - However, my point is that when it comes to ambiguous images, you'll have groups of images in the training dataset that contain similar features (along with some different ones), but have different labels, and groups that have the same labels but different features (along with some similar ones). That is another source of ambiguity, so "all the models were individually trained/fine-tuned on the same labels, so there's no ambiguity in their training" seems false.
> >   - Not only is this a source of ambiguity, it's a well-known one. And not only is it well-known, I think it's the one driving your results.
> >
> > I like the idea of investigating populations and your approach to experimentation. I also think the paper is well-written and visualized. However, I don't think you've found something nonobvious yet, and would encourage you to keep investigating. I agree with reviewer vUbw that more careful interpretation of similarity is necessary. I'll maintain my score.

---

### Official Review · Reviewer_vUbw · 2024-11-03

**Soundness:** 2
**Presentation:** 2
**Contribution:** 2
**Rating:** 3
**Confidence:** 3

**Summary:**

This paper assesses the disagreement among a population of artificial vision systems (1032 models) and compares it with the disagreement among a population of humans (42 human participants). Unlike previous works, populations of agents and humans are compared on a collective level, instead of an individual level. The paper aims to prove that factors that cause disagreement among AI systems coincide with the factors that cause human disagreement, at a population level.

**Strengths:**

The paper has the following (potentially) strong points:

1. The paper assesses the overlap between AI vision models and human disagreement on a collective/population level, rather than an individual level. This is an original approach as far as I know. The assumption is that by identifying patterns in how populations of AI models fail similarly to humans, training methods or architectures that handle difficult stimuli could be developed, and thus improve model robustness and interpretability. The proposed many-to-many comparison is something worth considering in the future, alongside already-established measures.

2. This study models the largest population (afaik) of artificial vision models, spanning 1032 AI models with various architectures, pretraining regimes and data. Such a population should provide a comprehensive view of collective disagreement. However, how each of these models influences the collective disagreement is not discussed enough, but could have been a point to add more value to the paper.

3. It aims to uncover and highlight common factors between humans and artificial models of vision that cause difficulty in object recognition.

**Weaknesses:**

This work presents the following weaknesses:

1. My first concern is related to the assumption from which the paper starts (L19) about the “ factors driving disagreement among AI systems are also causing misalignment between AI systems and humans perception” - why would that be the case?  It states that the current study challenges (L484) “the assumption present in prior work that disagreement among AI systems is unrelated to human visual processing”. But this assumption (L484) is not adequately founded, or at least not supported through the references provided which do not claim that disagreement between artificial models is unrelated to human visual processing. To reinforce, the initial assumption is not adequately discussed or supported by the correct references making it difficult to understand the motivation of the paper in the first place.


2. For a study comparing human and artificial visual systems, the authors might want to consider the body of literature that draws from neuroscience to better understand how convolutional neural networks (CNNs) could model early visual processing pathways [e.g. A Unified Theory of Early Visual Representations from Retina to Cortex (Lindsey et al., 2019); Spatial and Colour Opponency in Anatomically Constrained Deep Networks (Harris et al. , 2019)]. Such works aim to understand the similarities between human visual systems and artificial models at the lower level of neurons and how the functional and structural layouts of biological visual systems could better inform DNN architectures.

3. While the idea of comparing many to many is interesting and could add value on top of accuracy and one-to-one error consistency measures, the experimental setup seems to be (visually) ill-posed. For instance, the challenging examples are complex scenes, e.g. Figure 12, in which the label corresponds to just one small part of the scene. It should not be surprising that both humans and machines have difficulty in correctly identifying the target class in these cases. But it is not justified to use this as a basis to say that machines and humans are making mistakes in the same kind of way - it is much more nuanced than that.

4. While the assessment in Fig 6 aims to show the proportion of human-annotated top visual attributes, it is unclear on an instance level how and why humans and artificial models reach (dis)agreement. Take for example the cases where the model makes random kinds of predictions humans clearly would not. For example, Figure 3c is clearly not a roof tile, a scorpion, or a sandal - no human would guess any of those, although they could still be wrong of course.

**Questions:**

In light of the previous comments, I think the main actionable points are:
- the motivation of the paper needs to be reconsidered and clarified
- so does the conclusion and interpretation of results, in particular, I would recommend more carefully interpreting the similarities between humans and artificial models.

Further clarification is also needed on:
- Figure 1 -  the interpretation of the histograms for model and human agreement (“histograms along each axis reflect the proportion of images at each marginal agreement level”).  The caption states there is a positive correlation but does not state how this conclusion is reached.  Later on, Table 1 provides some values but the exact method for reaching those values is missing. Visually the histograms do not seem positively correlated, but again clarifying in text would be better.

- Details of the pretraining of each model, or at least grouped per family of models (maybe grouped by architecture type) used in this analysis would have been relevant. Also, further discussion and interpretation of results, again grouped per family of models could have added value to this paper. For example, how do different model architectures contribute to the level of disagreement?

- Again, for clarity,  it would be good to state clearly how the values for correlation between model agreement and the human behavioural measures (Table 1) are computed.

- Line 432 - What is this subset of selected models? Based on what criteria were these models selected?

- Regarding low-agreement images, it would be interesting to assess the factors that cause disagreement at certain levels of accuracy. Are these factors maintained, and what factors remain/are discarded as the acceleration of agreement occurs (as per L440-442)?

Finally, I think a section on the limitations of this study should be included. For example:
- the limited number of human participants might not reflect the full spectrum of human visual perception
- how does approximating perceptual abilities to population disagreement lead to overlooking specific, individual visual factors?
- is Fleiss’ Kappa the most suitable measure and are there any other agreement measures that could be explored instead?

---

> ### Author Response · Authors · 2024-11-12
> **Discussion of some key concerns**
>
> We value your thorough review and we're responding as timely as possible because the points you have made are important and we would like to discuss further.
>
> > **It should not be surprising that both humans and machines have difficulty in correctly identifying the target class in these cases. But it is not justified to use this as a basis to say that machines and humans are making mistakes in the same kind of way - it is much more nuanced than that.**
>
> We agree with this point. We are not saying that humans are making mistakes in the same kind of way, but humans and machines are potentially challenged in the same way by the same images. We think this is not obvious at the population level because all the models were individually trained/fine-tuned on the same labels, so there's no ambiguity in their training, but ambiguity and disagreement nonetheless arises. And that ambiguity and disagreement appears to be aligned with populations of humans.
>
> > **My first concern is related to the assumption from which the paper starts (L19) about the “ factors driving disagreement among AI systems are also causing misalignment between AI systems and humans perception” - why would that be the case?**
>
> While the actual notion of disagreement among a population of models has not been measured before our submission, it has been an explicitly stated assumption that the mistakes that AI models make are distinct from the mistakes that humans make. For instance, [Geirhos et al. (NeurIPS 2020)](https://arxiv.org/abs/2006.16736) make the points:
>
> "The consistency between CNNs and human observers, however, is little above what can be expected by chance alone—indicating that humans and CNNs are likely implementing very different strategies."
>
> "We conclude that there is a substantial algorithmic difference between human observers and the investigated sixteen CNNs: humans and CNNs are very likely implementing different strategies."
>
> “Cohen’s $\kappa$ for CNN-human consistency is very low for both models (`.068` for ResNet-50; `066` for CORnetS) compared to `.331` for human-human consistency.”
>
> Furthermore, [Geirhos et al. (NeurIPS 2018)](https://arxiv.org/abs/1808.08750) make the point:
>
> “Additionally, we find progressively diverging patterns of classification errors between humans and DNNs with weaker signals.”

---

> > ### Author Response · Authors · 2024-11-14
> >
> > > For a study comparing human and artificial visual systems, the authors might want to consider the body of literature that draws from neuroscience to better understand how convolutional neural networks (CNNs) could model early visual processing pathways
> >
> > Actually there's a growing body of work that shows that architecture does not play a significant role in modeling visual processing pathways.
> >
> > "We find that model scale and architecture have essentially no effect on the alignment with human behavioral responses, whereas the training dataset and objective function both have a much larger impact." \cite{muttenthaler2022human, conwell2024large}
> >
> > More recent work has found that many different architectures make very similar predictions \cite{conwell2024large} and discrimination along the dimension of architecture are not viable \cite{han2023system}. This was the original motivation for finding the 'disagreeable' stimuli, the variability of predictions amongst different models would help distinguish unique characteristics of potentially related to architecture. But to our surprise, as described in the introduction, the prediction variability among the model group was similar to the variability of human group.
> >
> > @article{muttenthaler2022human,
> >   title={Human alignment of neural network representations},
> >   author={Muttenthaler, Lukas and Dippel, Jonas and Linhardt, Lorenz and Vandermeulen, Robert A and Kornblith, Simon},
> >   journal={arXiv preprint arXiv:2211.01201},
> >   year={2022}
> > }
> >
> > @article{conwell2024large,
> >   title={A large-scale examination of inductive biases shaping high-level visual representation in brains and machines},
> >   author={Conwell, Colin and Prince, Jacob S and Kay, Kendrick N and Alvarez, George A and Konkle, Talia},
> >   journal={Nature Communications},
> >   volume={15},
> >   number={1},
> >   pages={9383},
> >   year={2024},
> >   publisher={Nature Publishing Group UK London}
> > }
> >
> > @inproceedings{han2023system,
> >   title={System identification of neural systems: If we got it right, would we know?},
> >   author={Han, Yena and Poggio, Tomaso A and Cheung, Brian},
> >   booktitle={International Conference on Machine Learning},
> >   pages={12430--12444},
> >   year={2023},
> >   organization={PMLR}
> > }

---

> > > ### Author Response · Authors · 2024-11-14
> > >
> > > > But it is not justified to use this as a basis to say that machines and humans are making mistakes in the same kind of way - it is much more nuanced than that.
> > >
> > > Could you clarify what is meant by nuanced here?
> > >
> > > > While the assessment in Fig 6 aims to show the proportion of human-annotated top visual attributes, it is unclear on an instance level how and why humans and artificial models reach (dis)agreement. Take for example the cases where the model makes random kinds of predictions humans clearly would not. For example, Figure 3c is clearly not a roof tile, a scorpion, or a sandal - no human would guess any of those, although they could still be wrong of course.
> > >
> > > Keep in mind that the percentage of models making those labels is far smaller (6% of models) for low agreement images than high agreement images (100% of models). So the model driven factors that lead to the labels of low agreement amongst models at the individual image level do not have consistency over a population of models by definition of the agreement metric. Based on the agreement metric, the prediction amongst models is very diverse and each model is making a prediction for reasons that are unique to that model amongst the 1032 models.

---

> > ### Comment · Reviewer_vUbw · 2024-11-25
> > **Follow-up discussion - score not changed**
> >
> > Thank you for addressing my comments.
> >
> > >While the actual notion of disagreement among a population of models has not been measured before our submission, it has been an explicitly stated assumption that the mistakes that AI models make are distinct from the mistakes that humans make. For instance, Geirhos et al. (NeurIPS 2020) make the points [...]
> >
> > I would like to start the discussion from here because I believe the authors still mix up **errors/mistakes** models and humans make and their **learning strategy**. The need for context in Geirhos et al 2020, as reviewer pHQG also mentioned, might be missing as they focus on an individual model vs human comparison. Hence, as I said in my initial review, it might be a useful addition. But this paper's results, as presented at this point do not carry significant information. I would encourage the authors to keep experimenting and trying to disseminate the contribution of how and what population disagreement (alone) provides value.
> >
> > > We agree with this point. We are not saying that humans are making mistakes in the same kind of way, but humans and machines are potentially challenged in the same way by the same images.
> >
> > This is exactly my point, how could you conclude your evaluation on a population level that models are challenged in the same way as humans? For example take your Fig 6 in which even though the trend looks similar, there is an almost 10% difference in background (and for pattern) between the models and the humans.
> >
> >
> > >This was the original motivation for finding the 'disagreeable' stimuli, the variability of predictions amongst different models would help distinguish unique characteristics of potentially related to architecture. But to our surprise, as described in the introduction, the prediction variability among the model group was similar to the variability of human group.
> >
> > My question here is how does that inform further studies? What should we do with these findings, which are neither convincing nor surprising? As I said in my example (Figure 12, in which the label corresponds to just one small part of the scene) it is very difficult to say why in this case models and humans made mistakes and whether it is for the same reason, or not.
> >
> > Discussing and interpreting the properties of images that elicit the most disagreement within the model population would've been the most interesting part (also related to my point on *nuanced*). What should we do about those challenging factors?
> >
> > And in line with this, let's discuss your other point...
> >
> > >Keep in mind that the percentage of models making those labels is far smaller (6% of models) for low agreement images than high agreement images (100% of models).
> >
> > This is not at all made clear in the paper.
> >
> > >So the model driven factors that lead to the labels of low agreement amongst models at the individual image level do not have consistency over a population of models by definition of the agreement metric. Based on the agreement metric, the prediction amongst models is very diverse and each model is making a prediction for reasons that are unique to that model amongst the 1032 models.
> >
> > This argument seems to contradict what you were saying earlier about the unique model (architecture), not impacting the prediction.
> >
> > To conclude my remarks, I will keep my score but do encourage the authors to clarify their motivation in line with available literature and to further experiment and provide interpretation for the driving factors causing disagreement and how this fits in, if at all, with the similarity between human and model perception.

---

> > > ### Author Response · Authors · 2024-11-26
> > >
> > > Thank you for the thoughtful discussion. We appreciate the comments as it improves our work and has generated follow-up analysis described below.
> > >
> > > Disagreement is an aggregate metric over a collection of models. The reasons for disagreement will not be uniform or unanimous among models just as they are not among a group of humans. We agree that would like to consider factors that cause or lead to disagreement. These causes can impact disagreement through in two inputs to the agreement calculation:
> > >
> > > 1) The aggregate behavior of the population of models
> > > 2) The images that the population of models operate over
> > >
> > > > This argument seems to contradict what you were saying earlier about the unique model (architecture), not impacting the prediction.
> > >
> > > To support and better understand the attributes that lead to agreement and disagreement, we have analyzed a counterfactual population by training 102 ResNet-50 models which were all trained on ImageNet with the only difference being the random seed used in initialization. This population produces nearly identical results for the ImageNet and ObjectNet dataset, (0.29 vs 0.29 correlation with MVT, 0.34 vs 0.41 correlation with ObjectNet). This is a very low diversity population of models that only vary over random seed which can replicate most of the results of the population of 1032 computer vision models which have been created over the past decade by the community. This indicates that most of the explainable variance in agreement/disagreement is not caused by variations in the model architecture nor differences in the training data.

---

> > > > ### Comment · Reviewer_vUbw · 2024-11-27
> > > >
> > > > Thanks for getting back with clarification of this experiment.
> > > >
> > > > > This is a very low diversity population of models that only vary over random seed which can replicate most of the results of the population of 1032 computer vision models which have been created over the past decade by the community. This indicates that most of the explainable variance in agreement/disagreement is not caused by variations in the model architecture nor differences in the training data.
> > > >
> > > > Very well. My question is what causes the variance if it's not model architecture or training data? You already pointed to references that discussed model architecture not having an impact (Muttenthaler 2022, Conwell 2024), so I do not understand if you are only confirming their finding or claiming this as novel.
> > > >
> > > > And going back to my main **concern** about your claim that humans and machines make mistakes in the same kind of way and why that is not true. Think, for example, of shortcut learning (Geirhos, 2020). The examples are very illustrative of the difference between machine and human strategies. DNNs correctly classify cows with a grass background, but not such much with an unusual background such as the sea. On the other hand, humans recognise cows based on their shape/pattern/colour and while they could still be influenced by context, it is not such an important factor.
> > > >
> > > > *Geirhos, R., Jacobsen, J.H., Michaelis, C., Zemel, R., Brendel, W., Bethge, M. and Wichmann, F.A., 2020. Shortcut learning in deep neural networks. Nature Machine Intelligence, 2(11), pp.665-673.*

---

> > > > > ### Author Response · Authors · 2024-11-27
> > > > >
> > > > > Thank you for continuing the discussion in improving this work.
> > > > >
> > > > > > And going back to my main concern about your claim that humans and machines make mistakes in the same kind of way and why that is not true.
> > > > >
> > > > > The minimum viewing time data we use from \cite{mayo2023hard} presents the minimum time necessary for a majority humans to correctly classify an image. We have done further analysis to decompose Figure 1 by images with different viewing times and find **disagreeable images take more time to recognize**. This trend can be seen to some degree in Figure 4, but the property becomes much more apparent simply by decomposing Figure 1 into different minimum viewing times.
> > > > >
> > > > > We would like to highlight that population disagreement is not the same concept as mistakes. Disagreement is more akin to image hardness, which explains why the most disagreeable images actually require more time to recognize rather than less. We show there is there is an intrinsic notion of disagreement that model populations capture as well as human populations. And with the new experiments with non-diverse populations, this intrinsic disagreement does not appear to be a function of variability on model architecture or variability of training data.
> > > > >
> > > > >
> > > > > @article{mayo2023hard,
> > > > >   title={How hard are computer vision datasets? Calibrating dataset difficulty to viewing time},
> > > > >   author={Mayo, David and Cummings, Jesse and Lin, Xinyu and Gutfreund, Dan and Katz, Boris and Barbu, Andrei},
> > > > >   journal={Advances in Neural Information Processing Systems},
> > > > >   volume={36},
> > > > >   pages={11008--11036},
> > > > >   year={2023}
> > > > > }

---

> > > > > > ### Comment · Reviewer_vUbw · 2024-11-27
> > > > > >
> > > > > > I believe you are not responding to my comment, but rather taking the discussion in a completely different direction. So before we proceed any further, could you please clarify what you understand by **disagreement** among model population?
> > > > > >
> > > > > > This confusion goes back to my request to motivate and convincingly explain the assumption that this paper starts from (L19-20).  In your very first reply, you gave the following answer:
> > > > > > >While the actual notion of disagreement among a population of models has not been measured before our submission, it has been an explicitly stated assumption that the mistakes that AI models make are distinct from the mistakes that humans make.

---

> ### Author Response · Authors · 2024-11-27
>
> Apologies for the confusion. Disagreement is defined per image in Equation 2. Based on the equation, it measures how "spread out" the predictions are between the population of models. The more spread out the predictions, the lower the agreement.
>
> > This confusion goes back to my request to motivate and convincingly explain the assumption that this paper starts from (L19-20). In your very first reply, you gave the following answer:
>
> The answer was pointing out that in the pair-wise error consistency metric from [Geirhos et al. (NeurIPS 2020)](https://arxiv.org/abs/2006.16736) was focused on the "trials where the decision makers agree do not provide much evidence for distinguishing between processing strategies. In contrast, the (few) errors of the decision makers are the most
> informative trials in this respect:."
>
> We do not mean to conflate that mistakes are the same as measures of agreement. Agreement is more analogous to prediction consistency. Our goal in L19-20 is to say that disagreement or lack of prediction consistency amongst a model population does not need to necessarily deviate from the lack of prediction consistency of a human population. We're open to suggestions on rewriting this to illustrate more clearly and thank you for the effort in discussion.

---

> > ### Comment · Reviewer_vUbw · 2024-11-28
> >
> > Thank you for the response. I will keep my score because my concerns regarding the motivation and assumptions in the paper were not solved. I appreciate the approach to investigating population disagreement. However, as I already said and as other reviewers also noted it is a well-known fact that ambiguous images that are known to be difficult for models can also be difficult for humans, but that can be for many different reasons. I encourage you to keep investigating and to more clearly interpret and present your results.

---

### Official Review · Reviewer_WTkd · 2024-11-04

**Soundness:** 2
**Presentation:** 2
**Contribution:** 1
**Rating:** 3
**Confidence:** 4

**Summary:**

The paper compares the collective behaviour of 1,032 AI vision systems with 42 humans in annotating images, investigating how various visual factors influence agreement levels. It highlights that images that are challenging for the AI systems often pose similar difficulties for humans. The paper suggests that there is an alignment in visual complexity across both groups. The study quantifies (dis)agreement among AI systems and compares the results with human annotations. Additional factors such as difficulty score, minimum viewing time, and specific visual properties are examined. This approach offers insights into common challenges shared by AI and human perception.

**Strengths:**

The comparison between model performance and human annotations is interesting and insightful.

**Weaknesses:**

- The paper is difficult to follow
- The motivation and contributions of the paper is not clear
- The paper lacks novelty, as it mainly consists of a comparison between the performance of machine learning models and human annotators. Reader may expect a novel methodology to be derived from these analyses.
- The paper lacks a discussion about the limitations and potential directions for future work

**Questions:**

- It is unclear why the authors concluded from Figure 1 alone that the stimuli causing the most agreement/disagreement among AI systems also cause the most agreement/disagreement among humans. Although the figure shows the agreement levels, it lacks specific information on the stimuli that contributed to such obtained outcomes
- In Table 1, what is the motivation behind comparing the models agreement with the human viewing time and the difficulty score?
- It is unclear why the authors concluded from Table 1 that ObjectNet is more challenging for both humans and the models?
- I would recommend to provide a correlation measure for Figure 5.
- Do you expect any bias in human annotations?
- In Figure 6, How did you determine the visual factors for the models?

---

### Meta-Review · Area_Chair_7WZo · 2024-12-17

**Metareview:**

After careful consideration of the six expert reviews and the subsequent author-reviewer discussion, I recommend rejecting this submission. While the paper presents an interesting analysis of population-level disagreement between AI vision systems and human perception, several fundamental concerns remain unresolved despite the authors' partial engagement with reviewer feedback.

The paper's central contribution examines how populations of AI models and humans show collective disagreement on certain visual stimuli. The authors suggest that this reveals an unexpected alignment between AI and human perception, particularly for challenging images. However, multiple reviewers questioned the novelty and significance of this finding. As Reviewer pHQG articulated, the observation that ambiguous images are difficult for both humans and models is well-known, and the correlation in disagreement patterns may simply reflect inherent image ambiguity rather than revealing meaningful mechanistic similarities.

The theoretical foundation and motivation of the work drew substantial discussions. While the authors cited prior work suggesting differences between human and AI visual processing strategies, reviewers noted this does not fully justify the paper's core assumption about disagreement patterns. Reviewer vUbw highlighted how the paper conflates different types of differences between human and machine vision - strategic differences versus simple classification errors. This distinction wasn't adequately addressed in the authors' responses.

The experimental methodology, while extensive in using over 1,000 AI models, raised concerns about interpretation. The authors' additional analysis showing similar results with 102 ResNet-50 models varying only in random seeds is interesting but, as pointed out in the discussion, does not clearly establish what drives the observed disagreement patterns if not architectural or training differences. The paper lacks a clear explanation of the causal mechanisms underlying the reported correlations.

Several reviewers also noted that the paper's practical implications remain unclear. As Reviewer tvpj emphasized, the results don't obviously point toward improvements in machine vision systems or deeper understanding of human vision. While the authors provided some responses about the value of studying edge cases, they didn't fully address how these findings could concretely advance the field.

The limited engagement with reviewer feedback is also concerning. While the authors had productive discussions with some reviewers, they left several comments unaddressed. The selective response pattern suggests that some issues with the work's motivation, interpretation, and significance remain unresolved.

While the paper presents an interesting analytical approach and extensive experimental work, the combination of unclear theoretical foundations, questionable novelty, and limited practical implications make it in an incomplete state for publication at ICLR 2025. For any future submission, the authors are recommended to focus on establishing clearer causal mechanisms, better distinguishing their findings from known phenomena, and more concretely demonstrating the work's significance for advancing either machine or human vision understanding.

**Additional Comments On Reviewer Discussion:**

During the rebuttal period, there was extensive discussion between reviewers and authors about several fundamental aspects of the paper. A key concern raised by multiple reviewers (vUbw, pHQG) centered on the paper's core assumption that disagreement among AI systems indicates an inherent limitation rather than reflecting human perceptual challenges. While the authors attempted to address this by citing prior work suggesting differences between human and AI visual processing strategies, these reviewers found the response less convincing, noting that the paper potentially conflates different types of differences between human and machine vision.

Several reviewers questioned the novelty and significance of the findings. Reviewer pHQG pointed out that the correlation between human and model disagreement on ambiguous images could simply reflect inherent image ambiguity rather than revealing meaningful mechanistic similarities. The authors responded by providing additional analysis using 102 ResNet-50 models with varying random seeds, showing similar disagreement patterns. However, this analysis, while interesting, did not fully address what drives these patterns if not architectural or training differences.

The authors engaged with concerns about experimental methodology and interpretation raised by Reviewer ihcH, clarifying that their human study included 2,647 participants rather than just 42, and explaining how edge cases could reveal important insights about visual intelligence. However, they did not address several other methodological concerns raised by other reviewers.

More fundamental questions about practical implications and future directions, raised particularly by Reviewer tvpj, received limited response. The authors' discussion of edge cases as “optical illusions for visual intelligence” did not fully satisfy concerns about how these findings could concretely advance either machine or human vision understanding.

The authors' engagement with reviewer feedback was selective, with some reviewers receiving detailed responses while others' concerns went unaddressed. This pattern of incomplete engagement suggests that several fundamental issues with the work's motivation, interpretation, and significance remain unresolved. In weighing these factors, the limited response to crucial theoretical and practical concerns, combined with the inability to convince reviewers of the work's novelty and significance, influenced the final rejection decision.

---

### Decision · Program_Chairs · 2025-01-22

Reject